# The gut microbiota promotes hepatic fatty acid desaturation and elongation in mice

Alida Kindt[1,10], Gerhard Liebisch [2], Thomas Clavel[3,4], Dirk Haller[4,5], Gabriele Hörmannsperger[4,5], Hongsup Yoon[4,5], Daniela Kolmeder[6], Alexander Sigruener[2], Sabrina Krautbauer[2], Claudine Seeliger[6], Alexandra Ganzha[6], Sabine Schweizer[6], Rosalie Morisset[6], Till Strowig [7], Hannelore Daniel[6], Dominic Helm [8], Bernhard Küster [8], Jan Krumsiek [1,9,11] & Josef Ecker[6]

Interactions between the gut microbial ecosystem and host lipid homeostasis are highly relevant to host physiology and metabolic diseases. We present a comprehensive multi-omics view of the effect of intestinal microbial colonization on hepatic lipid metabolism, integrating transcriptomic, proteomic, phosphoproteomic, and lipidomic analyses of liver and plasma samples from germfree and specific pathogen-free mice. Microbes induce mono-unsaturated fatty acid generation by stearoyl-CoA desaturase 1 and polyunsaturated fatty acid elongation by fatty acid elongase 5, leading to significant alterations in glycerophospholipid acyl-chain profiles. A composite classification score calculated from the observed alterations in fatty acid profiles in germfree mice clearly differentiates antibiotic-treated mice from untreated controls with high sensitivity. Mechanistic investigations reveal that acetate originating from gut microbial degradation of dietary fiber serves as precursor for hepatic synthesis of C16 and C18 fatty acids and their related glycerophospholipid species that are also released into the circulation.

[1] Institute of Computational Biology, Helmholtz Zentrum München, Neuherberg 85764, Germany. [2] Institute of Clinical Chemistry, Universitätsklinikum Regensburg, Regensburg 93053, Germany. [3] Functional Microbiome Research Group, Institute of Medical Microbiology, Universitätsklinikum Aachen, Aachen 52074, Germany. [4] ZIEL Institute for Food and Health, Technische Universität München (TUM), Freising 85354, Germany. [5] Ernährung und Immunologie, Technische Universität München (TUM), Freising 85354, Germany. [6] Ernährungsphysiologie, Technische Universität München (TUM), Freising 85354, Germany. [7] Research Group Microbial Immune Regulation, Helmholtz Centre for Infection Research, Braunschweig 38124, Germany. [8] Proteomics and Bioanalytics, Technische Universität München (TUM), Freising 85354, Germany. [9] German Center for Diabetes Research (DZD), Neuherberg 85764, Germany. [10] Present address: Department of Analytical Biosciences, Leiden Academic Centre for Drug Research, Leiden University, Leiden 2333, Netherlands. [11] Present address: Institute for Computational Biomedicine, Englander Institute for Precision Medicine, Department of Physiology and Biophysics, Weill Cornell Medicine, New York 10021, USA. These authors contributed equally: Alida Kindt, Gerhard Liebisch. Correspondence and requests for materials should be addressed to J.K. (email: jak2043@med.cornell.edu) or to J.E. (email: josef.ecker@tum.de)

The mammalian gut harbors highly complex microbial communities, referred to as the gut microbiota or microbiome when their genomes and surrounding environmental conditions are considered. Over the past decades, a number of mouse models have been used to study the gut microbiota and its relation to the host; these models include antibiotic-treated animals and germfree mice (GF) with or without colonization by specific microbial strains or defined consortia[1]. Previous studies have shown that the gut microbiota contributes to food processing; makes non-digestible nutrients available to the host; and provides important functions in host immunity, physiology, and metabolism[2,3]. The relationship between lipid metabolism and gut microbiota is important for host physiology and metabolic diseases. For example, fatty acids (FA) serve as precursors of signaling molecules and building blocks of a variety of lipids, including phospholipids (PL), which are major cell membrane constituents that contribute substantially to its flexibility[4,5]. Membranes undergoing rapid morphological changes contain high levels of poly-unsaturated fatty acids (PUFA), whereas membranes with biosynthetic functions are dominated by mono-unsaturated FA (MUFA)[6,7]. Numerous cellular processes, such as cell growth, cell differentiation, and organelle generation, require cellular de novo FA synthesis for cell membrane generation[8,9]. Fatty acid synthase (FASN) catalyzes the generation of palmitate (FA 16:0) from acetyl-CoA[10]. FA 16:0 can be further metabolized by stearoyl-CoA desaturase 1 (SCD1) and long chain fatty acid elongase 6 (ELOVL6)[11,12]. The precursors for PUFA are linoleic acid (FA 18:2 n-6) and α-linolenic acid (FA 18:3 n-3), which must be obtained from the diet and are precursors for both pro- and anti-inflammatory mediators[5]. A crucial organ in overall lipid homeostasis is the liver, playing critical roles in many metabolic diseases, particularly in type 2 diabetes[13].

To date, only a few studies have investigated the role of the gut microbiota in host lipid metabolism, primarily analyzing transcriptional profiles and the abundance of selected lipid species. In addition, some of the reported findings are partially contradictory. Bäckhed and colleagues have proposed that GF mice exhibit reduced hepatic FAs and adipose tissue triacylglyceride (TAG) synthesis compared to conventional mice[14,15], while Kimura et al. have recently proposed that the gut microbiota can suppress hepatic FA synthesis and lipid accumulation in adipose tissues[16]. A systematic study of altered lipid metabolic processes in GF mice is not yet available.

Thus, the aim of the present study was to establish a comprehensive overview of the hepatic processes affected by microbial colonization, with a particular focus on lipid metabolism. Multi-omics analyses were performed using samples obtained from GF and specific pathogen-free (SPF) C57BL/6 N mice. The hepatic transcriptome (32,308 transcripts), proteome (5875 proteins), phosphoproteome (5558 phosphopeptides), and the hepatic and plasma lipidome (525 lipid species of FAs, glycerophospholipids, sphingolipids, and sterols) were analyzed. Integrated and pathway-driven analysis identified that the gut microbiota triggers MUFA generation and PUFA elongation, leading to significant alterations in the acyl-chain profile of glycerophospholipids, including phosphatidylcholine (PC), -ethanolamine (PE), -inositol (PI), and PE-based plasmalogens (PE P). A gut microbiota-related molecular lipid signature was defined that could not only separate GF from SPF mice but also distinguish antibiotic-treated mice from untreated controls on the sole basis of their FA composition. Using in vivo stable isotope labeling experiments and dietary intervention strategies we showed that the short chain fatty acid (SCFA) acetate (FA 2:0) originating from gut microbial degradation of dietary fiber is a precursor for hepatic synthesis of long chain fatty acids and glycerophospholipids containing these fatty acids.

## Results

**Integrated transcriptome and proteome analyses.** To obtain a comprehensive view of hepatic processes influenced by microbial colonization, C57BL/6 N GF and SPF mice were fed a chow diet ad libitum and killed at 10 weeks of age. Liver and plasma samples were split for multi-omics analyses. Liver micro-architecture and weight were similar in SPF and GF mice (Supplementary Fig. 1). We investigated the hepatic transcriptome of C57/BL6N mice using microarrays. In total, 41,174 probes were analyzed, of which 32,308 could be mapped to transcripts and 21,461 had unique Entrez gene IDs. After correcting for multiple testing by controlling the false discovery rate at 0.05, 7769 probes were significantly differentially expressed (3937 genes), with fold changes up to 14.3 in comparisons between GF and SPF (Fig. 1a; Supplementary Data 1). The five most differentially expressed genes were $Tbc1d8$ [fold change (FC), $-1.8$; $p = 3.43 \times 10^{-12}$; unknown function]; $Igh$-$vj558$ (FC, 3.81; $p = 5.50 \times 10^{-12}$; unknown function), $Tpmt$ (FC, $-3.65$; $p = 2.33 \times 10^{-11}$; methylation of thiopurine compounds), $Itk$ (FC, 5.40; $p = 2.39 \times 10^{-11}$; tyrosine kinase), and $Pparg$ (FC, 2.05; $p = 2.65 \times 10^{-11}$; nuclear receptor involved in lipid metabolism). Pathway enrichment analysis revealed a number of metabolic pathways significantly enriched in GF mice, including those involved in drug metabolism, in agreement with a previous study[17], and lipid metabolism (4 of 10 pathways; Fig. 1b).

Although hepatic proteome profiling of GF mice has been reconstructed by others through a modeling approach[17], the full liver proteome of GF animals has not been measured. Mass-spectrometry-based analysis of liver samples was performed using liquid chromatography (LC) coupled to high resolution (HR) tandem mass spectrometry (MS/MS). Proteins were identified with the UniProtKB mouse database and considered as present if available in all samples per group. The mouse liver proteome comprised 5875 measured proteins that corresponded to 5630 unique proteins that could be mapped. After correcting for multiple testing by controlling the false discovery rate (FDR < 0.05), 469 significantly different peptides (455 unique proteins) were identified between GF and SPF animals (Fig. 1c; Supplementary Data 2), with fold changes up to 3.7. The five most differentially expressed proteins were L3HYPDH (FC, 1.59; $p = 7.32 \times 10^{-8}$; trans-3-hydroxy-L-proline dehydration), CYP3A11 (FC, 2.03; $p = 3.80 \times 10^{-7}$; cytochrome P450 protein), EPHX1 (FC, 1.49; $p = 3.59 \times 10^{-7}$; epoxide hydrolysis), SCD1 (FC, 2.68, $p = 8.23 \times 10^{-7}$; palmitate desaturation), and ACSF2 (FC, 1.36; $p = 1.26 \times 10^{-6}$; fatty acyl-CoA biosynthesis). Again, several significant proteins could be assigned to specific metabolic processes, including linoleic acid and retinol metabolism (Fig. 1d).

We combined the proteome and transcriptome data using gene names in order to identify genes that were up- or downregulated in both omics layers. This left 4843 common, uniquely mapped genes (Fig. 1i). In total, 2822 entities (~58%) were co-regulated, of which 984 were consistently higher in SPF (Fig. 1e, Q2) and 1838 consistently higher in GF (Fig. 1e, Q4). Only 5.4% of the genes were significantly differentially regulated (Q2, 71; Q4, 80). Functional enrichment analysis of the genes present in both the proteome and transcriptome showed that commonly upregulated genes (Q2) in SPF mice were enriched in 5 of 8 (~63%) GO biological processes involved in lipid metabolism (Fig. 1f). Downregulated genes (Q4) shared no enriched GO annotations. Significantly changed key enzymes of lipid metabolism differentially expressed in both omics included ACLY, FASN, SCD1, and ELOVL6, which catalyze the synthesis of saturated fatty acids (SAFA) and MUFA from acetyl-CoA; and SC5D and LSS, which are involved in cholesterol biosynthesis.

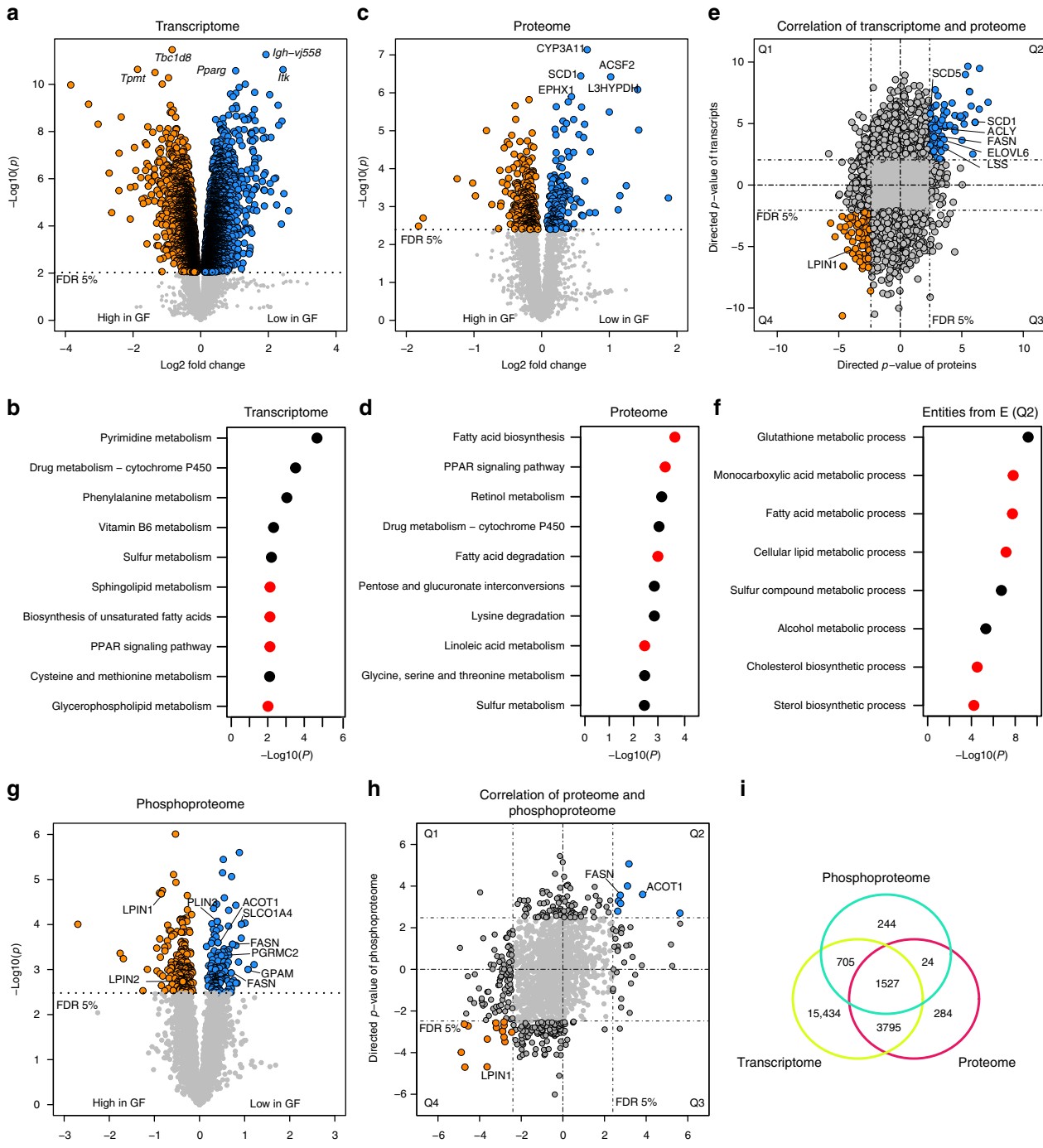

**Fig. 1** Transcriptomic, proteomic, and phosphoproteomic analyses of liver samples from SPF and GF mice. **a** Transcriptome data analyzed by *t*-tests, (*n* = 6/6). **b** KEGG pathway enrichment on transcriptome data. **c** Proteome data, analyzed by *t*-tests, (*n* = 5/5). **d** KEGG pathway enrichment in proteome data. **e** Transcriptome–proteome correlation. **f** GO enrichment analysis for biological processes of Q2 from **e**. **g** Phosphoproteome analyzed by *t*-tests, (*n* = 5/5). **h** Proteome–phosphoproteome comparison. **i** Venn diagram showing the overlap of detected genes, proteins, and phosphoproteins. **e**, **h** Q1, Q2, Q3, and Q4 mark the quadrants of the correlation plots; directed *p*-values are defined as–log$_{10}$ (*p*) times the direction of the effect. **b**, **d**, **f** Red dots indicate lipid-related pathways. FDR: false discovery rate, GF: germfree, SPF: specific pathogen-free

**Phosphoproteome profiling.** Protein phosphorylation plays an important role in the activation and deactivation of enzymes. Thus, we conducted a phosphoproteome analysis of liver samples from GF and SPF mice by LC-MS/HR-MS. We identified a total of 5558 phosphopeptides corresponding to 2500 unique proteins (Supplementary Data 3). For 1551 of these phosphopeptides, we could also detect the corresponding proteins, and 1527 of the phosphopeptides were detected in all three omics, i.e.,

transcriptome, proteome, and phosphoproteome (Fig. 1i). We found that 363 proteins had a significantly different phosphorylation status between GF and SPF animals (Fig. 1g). Of these, eight proteins were involved in lipid metabolism: ACOT1 (hydrolysis of acyl-CoA), FASN, GPAM (glycerolipid synthesis), LPIN1/2 (phosphatidic acid phosphatases), PGRMC2 (steroid binding), PLIN3 (lipid droplet coat protein) and SLCO1A4 (bile acid transport). After combining protein and phosphoprotein

**Table 1 Summary of lipids quantified in plasma and liver samples from GF and SPF animals using mass-spec–based methods**

| Lipid category | Analyzed species | Species detected in liver | Species detected in plasma | Lipid class | Method |
|---|---|---|---|---|---|
| Total FA | 33 | 27 | 21 | Total fatty acids (FA) | GC-MS |
| Glycero-phospholipids | 377 | 222 | 133 | Phosphatidylcholine (PC)[a] | [a]ESI-MS-MS |
| | | | | Phosphatidylethanolamine (PE)[a] | [b]LC-MS/MS |
| | | | | Phosphatidylserine (PS)[a] | |
| | | | | Phosphatidylglycerol (PG)[a] | |
| | | | | Phosphatidylinositol (PI)[a] | |
| | | | | PE-based plasmalogens (PE P)[a] | |
| | | | | Lyso-PC (LPC)[a] | |
| | | | | Phosphatidic acid (PA)[bc] | |
| | | | | Lyso-PA (LPA)[bc] | |
| | | | | Lyso-PG (LPG)[bc] | |
| | | | | Cardiolipin (CL)[bc] | |
| | | | | Bis(monacylglycero)phosphate (BMP)[bc] | |
| Sphingolipids (SL) | 77 | 42 | 33 | Sphingomyelin (SM)[a] | [a]ESI-MS-MS |
| | | | | Ceramide (Cer)[a] | [b]LC-MS/MS |
| | | | | Dihexosylceramide (Hex2Cer)[bc]; | |
| | | | | Hexosylceramide (HexCer)[bc] | |
| | | | | Sphinganine (SPA) | |
| | | | | Sphingosine (SPH)[bc] | |
| | | | | Phytosphingosine (Phyto-SPH) | |
| | | | | Sphingosine-1-phosphate (S1P)[bc] | |
| | | | | Sphingosylphosphorylcholine (SPC)[bc] | |
| Sterols | 38 | 23 | 23 | Free Cholesterol (FC) | ESI-MS/MS |
| | | | | Cholesterylester (CE) | |

ESI: electrospray ionization, GC: gas chromatography, LC: liquid chromatography, MS/MS: tandem mass spectrometry
[c]Not analyzed in plasma

abundances using the Uniprot KB identifier, we could identify 24 unique proteins (Q1, 1; Q2, 8; Q3, 1; Q4, 14), including ACOT1 (Q2), FASN (Q2), LPIN1 (Q4) that differed significantly in protein abundance and phosphorylation status (Fig. 1h).

**Quantitative lipidomics**. To test whether the changes observed for enzymes involved in FA and lipid metabolism lead to altered lipid levels in GF mice, we performed a comprehensive, quantitative lipidome analysis. Plasma and liver levels of 525 lipid species belonging to 20 classes from the categories of FAs, glycerophospolipids, sphingolipids, and sterols were analyzed by quantitative methods using gas chromatography coupled to mass spectrometry (GC-MS), direct flow injection tandem MS (FIA-MS/MS) and LC coupled to tandem MS (LC-MS/MS; Table 1). Lipidomic analyses were performed on the same samples used for the other omics analyses (GF: $n = 6$; SPF: $n = 6$) and another set of samples from an additional experiment with larger numbers of mice per group (GF: $n = 12$; SPF: $n = 14$) housed under the same conditions. The data from the two studies were measured in separate batches and processed separately.

Lipid species were considered significantly different between GF and SPF mice if their $p$-value passed the multiple-testing correction threshold (FDR < 0.05) and showed the same direction of regulation in both experiments. In total, 16 lipid species fulfilled these criteria in liver and 15 in plasma samples (Table 2; Fig. 2 & Supplementary Data 4). Most importantly, we found a systematic shift from mono-unsaturated to polyunsaturated lipid species between SPF and GF mice. Levels of palmitoleic acid (FA 16:1 $n$-7) (Fig. 2a, c) and glycerophospholipid levels containing monounsaturated acyl chains such as phosphatidylcholine (PC) 34:1 and PC 36:1 (liver only) (Fig. 2b, d) were as much as 1.5 fold higher in samples from SPF mice. In contrast, concentrations of the polyunsaturated FA and glycerophospholipid species arachidonic acid (FA 20:4 $n$-6; plasma only), docosahexaenoic acid (DHA, FA 22:6 $n$-3; liver only), phosphatidylinositol (PI) 40:6,

plasmalogen (PE P) 16:0/20:4, and PE 16:0/20:4 were higher in GF animals (Fig. 2a, c & Supplementary Data 4). The only exceptions were dihomo-γ-linoleic acid (FA 20:3 $n$-6; liver only) (Fig. 2a) and lyso-PC (LPC) 20:3 (Supplementary Data 4), whose levels were lower in GF mice. Sphingolipid species were not affected by gut microbiota colonization (only analyzed in liver samples; Supplementary Data 4). Of the sterols, cholesteryl ester (CE) 20:4 levels particularly were altered (higher in GF; Supplementary Data 4).

PC (37%), phosphatidylethanolamine (PE, 19%), and FC (11%) were the major lipid classes in liver (Fig. 2e). Sphingolipids represented only a minor fraction (~2%) of liver lipids (Fig. 2f). Plasma lipids were dominated by PC (27%), CE (54%), and FC (10%) (Fig. 2g). The total concentrations per lipid class did not differ between the groups in liver and plasma samples (Fig. 2e, g).

These results are further supported by the finding that Oligo-MM[12] mice show similar differences of the liver and plasma FA profile if compared to GF mice as SPF mice (Supplementary Figure 2A, B). Oligo-MM[12] mice are housed in isolators within a gnotobiotic environment as GF mice, they harbor a community of 12 microbial strains representing members of the major bacterial phyla in the murine gut including *Bacteroidetes* and *Firmicutes*[18].

In summary, these data show that the gut microbiota affects particular FA and glycerophospholipid species profiles in liver and plasma. Mice containing gut microbiota have higher proportions of MUFA-containing lipids, while lipids from GF mice were dominated by SAFA and PUFA.

**Reconstruction of fatty acid metabolism from omics data**. We hypothesized that GF mice differ from SPF mice in their FA desaturation and elongation capacities, leading to altered hepatic glycerophospholipid acyl-chain profiles. Thus, we reconstructed the FA metabolic pathways and integrated the phospho-, proteome, and transcriptome for a systematic overview on hepatic lipid metabolic processes (Fig. 3a, b). We observed striking

**Table 2 Verified lipid species that significantly differ in plasma and liver samples between GF and SPF mice**

| Lipid species | Lipid class | Matrix | Log FC Exp.1 GF = 6; SPF = 6 | Log FC Exp 2 GF = 12; SPF = 14 | *p* Exp. 1 | *p* Exp. 2 |
|---|---|---|---|---|---|---|
| FA16:1 *n*-7 | FA | Liver | 1.39 | 0.31 | 0.014 | 0.019 |
| FA20:3 *n*-6 | | | 0.48 | 0.18 | 0.003 | 0.039 |
| FA22:6 *n*-3 | | | −0.40 | −0.18 | 0.011 | 0.011 |
| FA16:1 *n*-7 | FA | Plasma | 1.41 | 0.36 | 0.013 | 0.014 |
| FA20:4 *n*-6 | | | −1.09 | −0.41 | 0.021 | 0.039 |
| PC 32:0 | PC | Liver | −0.36 | −0.22 | 0.006 | 0.018 |
| PC 34:1 | | | 0.98 | 0.34 | 0.009 | 0.029 |
| PC 36:1 | | | 0.89 | 0.33 | 0.014 | 0.002 |
| PC 38:2 | | | 1.47 | 0.52 | 0.019 | 0.001 |
| PC 38:3 | | | 0.61 | 0.36 | 0.011 | 0.002 |
| PC O-36:1 | | | 0.56 | 0.18 | 0.006 | 0.010 |
| PC O-36:4 | | | −0.55 | −0.24 | 0.030 | 0.017 |
| PC 34:1 | PC | Plasma | 1.17 | 0.35 | 0.004 | 0.004 |
| PC 38:3 | | | 0.55 | 0.17 | 0.024 | 0.006 |
| PC O-38:4 | | | −0.60 | −0.36 | 0.017 | 0.014 |
| LPC 20:3 | LPC | Liver | 1.18 | 0.47 | 0.005 | 0.002 |
| LPC 20:3 | LPC | Plasma | 1.14 | 0.30 | 0.006 | 0.000 |
| PE 32:0 | PE | Liver | −0.41 | −0.55 | 0.041 | 0.001 |
| PE 38:5 | PE | Plasma | 0.31 | 0.21 | 0.030 | 0.039 |
| PE P-18:0/20:4 | PE P | Liver | −0.50 | −0.17 | 0.011 | 0.021 |
| PE P-16:0/20:4 | PE P | Plasma | −0.91 | −0.32 | 0.013 | 0.003 |
| PE P-16:0/22:6 | | | −1.20 | −0.45 | 0.010 | 0.004 |
| PE P-18:0/20:4 | | | −0.95 | −0.26 | 0.032 | 0.039 |
| PE P-18:0/22:6 | | | −1.33 | −0.45 | 0.029 | 0.003 |
| PI 36:2 | PI | Liver | −1.16 | −0.91 | 0.001 | 0.000 |
| PI 40:6 | | | −0.92 | −0.88 | 0.005 | 0.000 |
| PI 36:4 | PI | Plasma | 0.23 | 0.38 | 0.038 | 0.000 |
| PI 40:6 | | | −0.75 | −0.76 | 0.003 | 0.000 |
| CE 20:4 | CE | Liver | −0.65 | −0.29 | 0.014 | 0.007 |
| CE 20:3 | CE | Plasma | 0.95 | 0.32 | 0.017 | 0.018 |
| CE 20:4 | | | −0.72 | −0.35 | 0.017 | 0.006 |

FC: fold change, Exp.: experiment, PC: *O* alkyl-acyl-PC. For further abbreviations see Table 1

differences between GF and SPF mice in three specific FA conversion steps: (I) De novo synthesis of FA from acetyl-CoA by ACAC A/B and FASN (Fig. 3a); (II) Delta-9 desaturation of palmitate (FA16:0) to palmitoleate (FA16:1 *n*-7) by SCD1 (Fig. 3a); (III) Elongation of FA18:3 *n*-6 to FA20:3 *n*-6 by ELOVL5 (Fig. 3b), with higher values for SPF in all steps. To support (II) and (III) we correlated the FA product/precursor ratios to the corresponding mRNA or protein abundance. As shown in Fig. 3d, SCD1 expression strongly correlated with the 16:1 *n*-7/FA16:0 ratio (mRNA: $R = 0.93$; $p < 0.00002$; protein: $R = 0.94$; $p < 0.00005$) and ELOVL5 expression strongly correlated with the 20:3 *n*-6/18:3 *n*-6 ratio (mRNA: $R = 0.79$; $p < 0.002$; protein: not detected). Protein and transcript abundances of sterol regulatory element binding protein (SREBP) 1 C and liver X receptor (LXR) α, two major transcription factors controlling many genes of FA synthesis and PUFA metabolism, did not differ (Fig. 3c). However, we observed different expression levels of lipin (LPIN) 1/2 and the transcription factors peroxisome proliferator activated receptor (PPAR) α and PPAR γ coactivator 1α (PPARGC1A) (Fig. 3c), which act in a synergistic way for SREBP-1c activation[19–21].

We also found that the reaction ratios (product/precursor levels) of the FA fraction correlate highly with the corresponding desaturation and elongation indices of PC and LPC species (Fig. 3e), indicating that the altered FA metabolic processes are clearly reflected in the acyl-chain profiles of glycerophospolipids.

modulates lipid metabolic processes (Fig. 3a, b; Reactions I, II, and III), we manipulated the gut microbial ecosystem of SPF animals via short-term antibiotic treatment, including ampicillin (A; 1 g/L; *n* = 6), vancomycin (V; 0.25 g/L; *n* = 6), metronidazole (M; 1 g/L; *n* = 5), or a combination of V (0.25 g/L) and M (1 g/L, *n* = 6). Hepatic FA species were quantified using GC-MS. To evaluate the effects on host FA metabolism, we first calculated the FA signature that separates GF from SPF animals with a minimal number of FA species using the data from experiments 1 and 2 (GF, total *n* = 18; SPF, total *n* = 20). The selected FAs were then used to calculate a classification score. The FA signature and classification score were tested on data obtained from a third experiment (GF, *n* = 5; SPF, *n* = 6; conditions identical to those in experiments 1 and 2) and the antibiotics experiment.

Classification scoring was effective in clearly separating not only GF from SPF animals with high specificity and sensitivity using FA 16:0, FA 20:3 *n*-6, FA 20:4 *n*-6, and FA 22:6 *n*-3 (Fig. 4a, b), but also separating mice receiving different antibiotics from those in the untreated control group (Fig. 4c). M alone or in combination with V had the lowest classification score and thus the most profound effect on those FA determined to be representative of the GF condition. Messenger RNA expression levels of *Scd1* and *Elovl5* clearly followed the trend of the classification score with low expression levels in samples with low scores (Fig. 4d). These results clearly support that gut microbiota affect FA metabolism, particularly FA species generated by SCD1 and ELOVL5 (Fig. 3a, b; reactions II and III).

**Composite classification score based on fatty acid profiles**. To provide further evidence that the gut microbiota specifically

**Gut microbiota composition analysis**. To determine the effect of antibiotics on gut microbiota diversity and composition, we

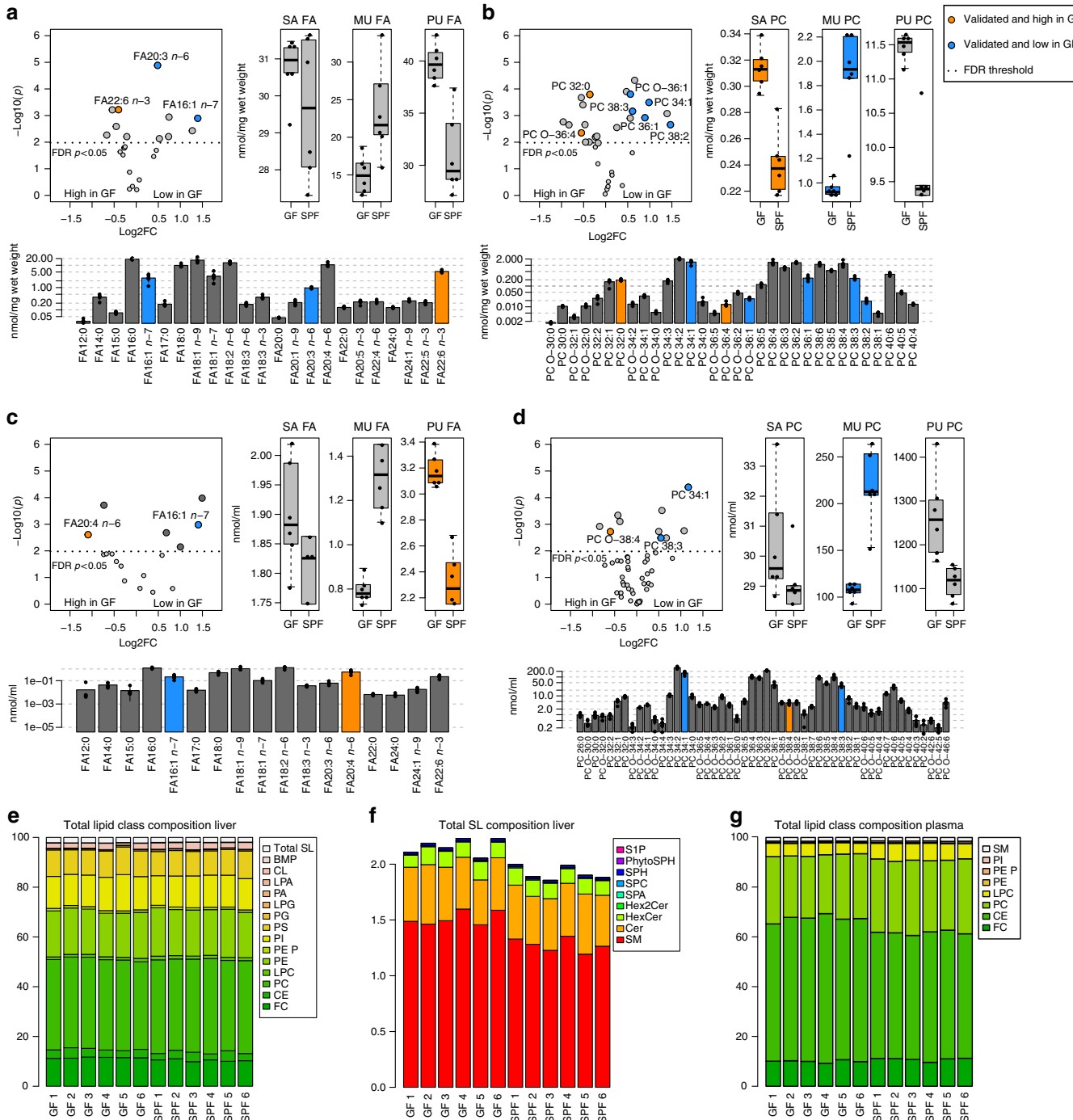

**Fig. 2** Quantitative lipidome analyses of liver and plasma from GF and SPF mice. Data from experiment 1 are shown ($n = 6/6$). Candidates verified in experiment 2 (SPF: $n = 14$; GF: $n = 12$) are displayed in blue (high in SPF) or orange (high in GF). **a** Total fatty acids (FA) in liver. **b** PC species in liver. **c** Total FA in plasma. **d** PC species in plasma. **a–d** Volcano plots: significance and log$_2$ fold change in individual lipid species; boxplots: concentrations of saturated (SA), monounsaturated (MU), and polyunsaturated (PU) lipid species in GF and SPF mice; barplots: molecular lipid species profile of GF animals. In boxplots the thick lines represent the medians, the upper and lower lines of the boxes show the 25 and 75% quartiles and the whiskers are 1.5 times the interquartile range of the data. Error bars in the bar plots show the standard deviation. **e** Lipid class composition of liver samples, experiment 1. **f** Sphingolipid composition of liver samples, experiment 1. **g** Lipid class composition of plasma, experiment 1. GF: germfree, PC: phosphatidylcholine (diacyl), PC: O phosphatidylcholine (alkyl-acyl), SPF: specific pathogen-free, for other abbreviations, see Table 1

analyzed V3-V4 amplicons of 16 S rRNA genes by high-throughput sequencing. We were unable to obtain 16 S rRNA gene amplicons in sufficient quality and amount from the cecal content of A-treated mice. For all other samples, we obtained a total of 376,071 sequences (17,094 ±3179 sequences per sample) representing a total of 172 operational taxonomic units (OTU)

(99±49 OTUs per sample). All antibiotics affected alpha-diversity (Fig. 4e, f), with particularly drastic effects by V and the combination of V and M (VM) compared to M alone, especially when considering most dominant species by taking into account evenness via calculation of Shannon effective counts (Fig. 4f). Beta-diversity analysis revealed a significant clustering of V and

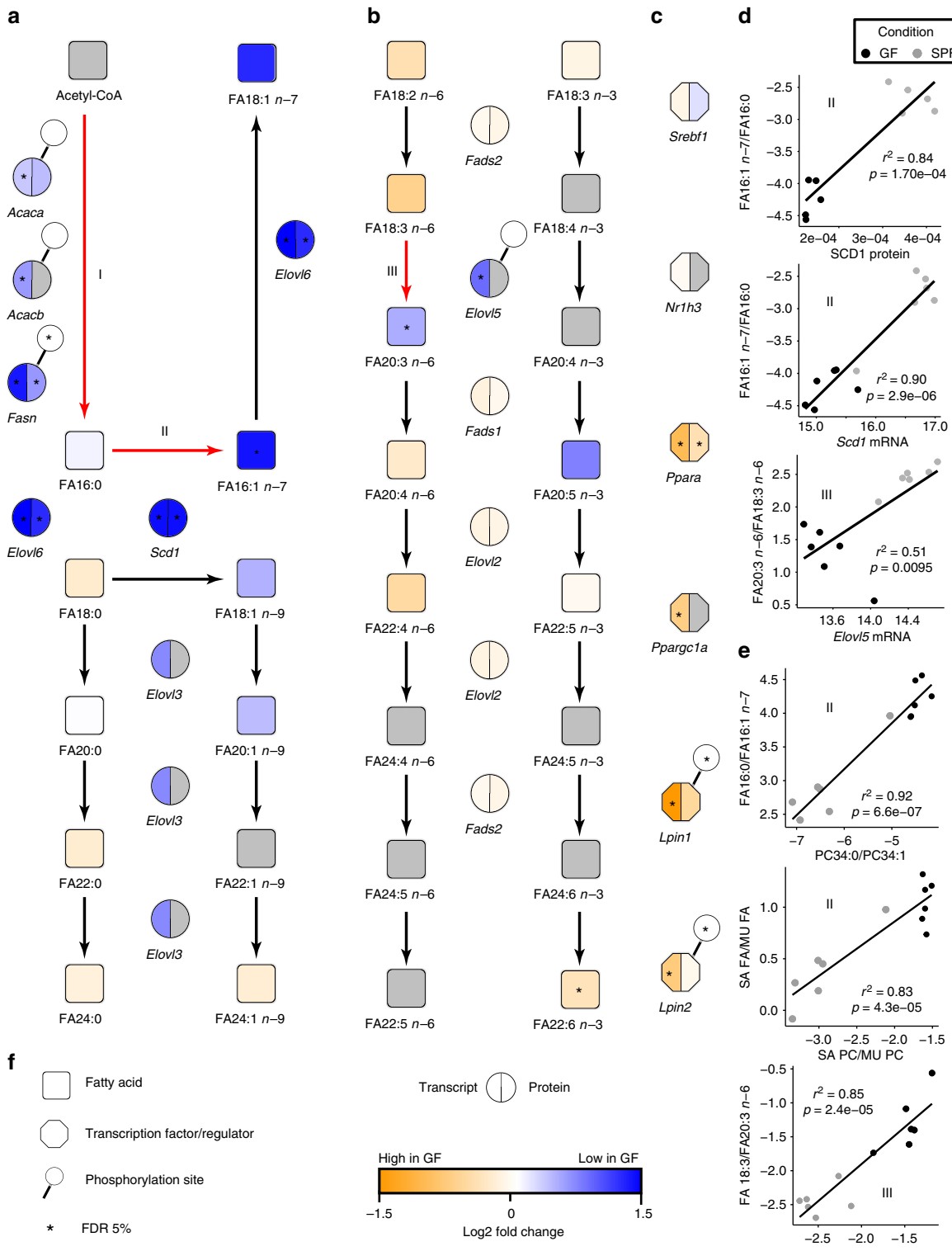

**Fig. 3** Reconstruction of FA metabolic pathways from multi-omics data and FA-glycerophospholipid species correlations. **a** Cellular de novo synthesis of palmitate, elongation to saturated and desaturation to monounsaturated FA. **b** Metabolism of *n*-3 and *n*-6 PUFA. (I–III) indicate the key processes altered in GF compared to SPF mice. **c** Transcription factors and regulators controlling **a** and **b**. **d** Correlation of product to precursor ratios with the appropriate gene or protein expression for reactions (II) and (III). **e** Correlation of FA species ratios with corresponding PC or LPC ratios for reactions (II) and (III). **f** Legend for the figure highlighting the strength and direction of the association for all molecular entities shown. FA: fatty acid, LPC: Lyso-PC, MUFA: monounsaturated fatty acids, PC: phosphatidylcholine, PUFA: polyunsaturated fatty acids, SAFA: saturated fatty acids, for other abbreviations, see Table 1

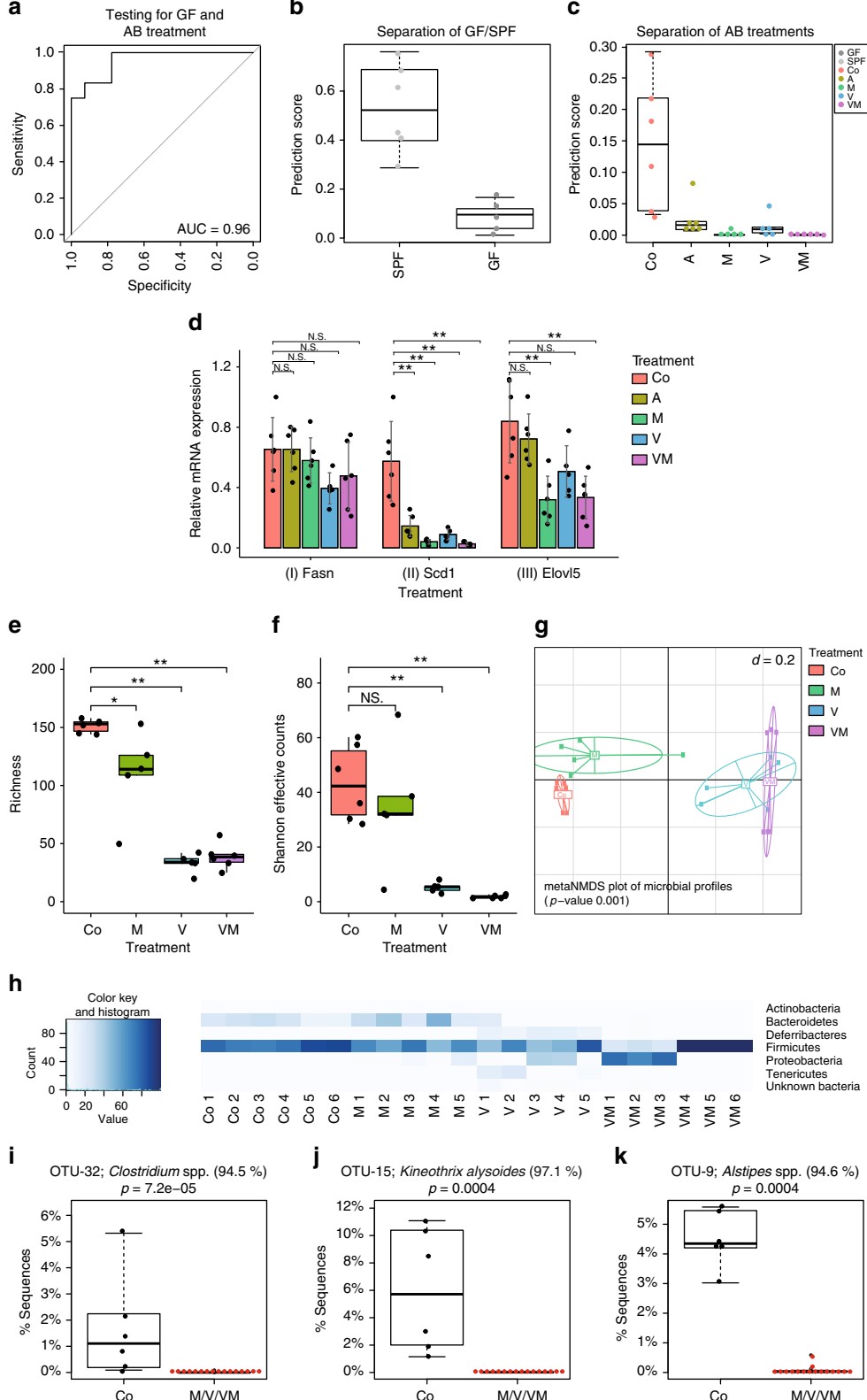

VM samples, which were not separated from one another but were most distant from the control group (Fig. 4g). The most dominant phylum, with ~65% relative abundance across all samples, was *Firmicutes* (Fig. 4h). In contrast to M, V almost completely eradicated *Bacteroidetes* (except for sample V1), while *Deferribacteres* were exclusively present in animals treated with V alone (13% relative sequence abundance on average). Next, we

asked if there were associations between the classification score and specific OTUs or gene expression in the antibiotics data. The score differed strongly between the control and antibiotic treatments but only weakly between the different AB treatments. Thus, we investigated which of the dominant OTUs (>1% relative abundance) were eradicated most effectively by the antibiotics used. The most significantly affected OTUs were OTU-32,

**Fig. 4** Effect of antibiotics on hepatic FA metabolism and association of cecal microbial species with FA metabolic consequences. SPF mice were treated with ampicillin (A, $n = 6$), metronidazole (M, $n = 5$), vancomycin (V, $n = 6$) or a combination of V and M (VM, $n = 6$). **a** Classification sensitivity and specificity of the calculated score separating GF and antibiotic-treated from the SPF mice. **b** Score for GF and SPF mice (GF: $n = 5$, SPF: $n = 5$) from a third experiment to validate the classification score. **c** Score for antibiotic-treated and untreated mice. **d** Relative mRNA expression of *Fasn*, *Scd1*, and *Elovl5*, mean and standard deviation are shown. **e** Alpha-diversity analysis shown as richness counts, and **f** Shannon effective counts. Samples from A-treated mice reproducibly generated too few sequences and thus could not be included in 16 S rRNA gene amplicon analysis. **g** Multidimensional scaling showing differences in the phylogenetic makeup of microbiota between samples (β-diversity) based on general UniFrac distances. **h** Microbiota composition at the phylum level. **i-k** Most significantly different OTUs after a Kruskal–Wallis analysis in untreated compared to antibiotics treated mice. N.S.: not significant, OTU: operational taxonomic unit. In boxplots the thick lines represent the medians, the upper and lower lines of the boxes show the 25 and 75% quartiles and the whiskers are 1.5 times the interquartile range of the data. In barplots the error bars show the standard deviation

*Clostridium* spp. (*Firmicutes*), OTU-15, *Kineothrix alysoides* (*Firmicutes*) and OTU-9, *Alistipes* spp. (*Bacteroidetes*), which together accounted for ~12% of all detected sequences in untreated SPF mice (Fig. 4i–k).

**Luminal acetate as precursor for hepatic lipid synthesis.** Members of *Bacteroidetes* and *Firmicutes* including *Kineothrix alysoide*s (OTU-15) degrade dietary polysaccharides from fiber to short chain fatty acids (SCFA) including acetate (FA 2:0), which reach the liver via the portal vein[22,23]. Portal blood FA 2:0 levels are ~0.5 mM (FA 2:0/FA 3:0/FA 4:0: ~50/3/1) in mice, rats, and humans[24,25].

To ask whether FA 2:0 originating from the gut lumen is a precursor for synthesis of long chain FA in the liver, mice were supplied with different concentrations of $^{13}$C-labeled FA 2:0 via oral gavage. After 4 h liver and plasma FA 16:0 were analyzed for $^{13}$C-enrichment using GC-MS. Administration of $^{13}$C-FA 2:0 clearly enhanced isotopologues $M_2$-$M_6$ of FA 16:0 in a dose-dependent way in both liver and plasma (Fig. 5a, b) resulting in an increase of the fractional abundances of $M_2$–$M_6$ and decrease of monoisotopic $M_0$ (Fig. 5c, e). $^{13}$C-Acetate also dose-dependently elevated the fraction of newly synthesized FA 16:0 in the liver 4 h after oral gavage from ~4 to 30 % demonstrating a stimulation of the hepatic FA *de novo* synthesis by gut-derived acetate (Fig. 5d).

To demonstrate that differences in the lipid profiles of GF and colonized mice (Fig. 2, Supplementary Figure 2), particularly altered MUFA contents, can be associated with microbial FA 2:0 production in the gut, we next manipulated *Bacteroidetes* and *Firmicutes* as major SCFA producers. SPF mice received a combination of vancomycin and metronidazole (VM; as described before) for 2 days (time-point 2, TP2), before they obtained a regular chow diet without antibiotics for additional two (TP4) or 10 days (TP 14). At TP2 + VM gut microbial composition and diversity dropped (α- and β-diversity; Fig. 5f, g, h). *Bacteroidetes* and *Firmicutes* were almost completely eliminated (Fig. 5i) and portal vein FA 2:0 concentrations were reduced twofold (Fig. 5j). After removal of the antibiotics (TP4 and TP14), the gut microbial ecosystem recovered and baseline FA 2:0 levels (TP0) were reached. Importantly, alterations of liver and plasma MUFA (16:1 *n-7*, FA 18:1 *n-9*, 18:1 *n-7*), MUPC (PC 34:1, PC 36:1) and PUFA (20:3 *n-6*, 22:6 *n-3*) levels clearly followed this trend (Fig. 5k, l). These results indicate a direct relation of liver and plasma FA levels to gut microbial SCFA producers and portal blood FA 2:0 levels.

Chow diet used in these experiments comprises polysaccharides from a grain-soybean-based crude fiber extract (5%) that can be depolymerized and subsequently fermented by gut microbiota to SCFA. To demonstrate that hepatic synthesis of MUFA depends on gut microbial degradation of dietary fiber, GF and SPF mice were fed with an experimental control diet with carbohydrate, fat and protein content comparable to chow, but with 5% purified cellulose instead of crude fiber. Refined cellulose

is practically non-degradable and -fermentable by gut microbiota leading to markedly reduced SCFA levels in portal blood[26–28]. In contrast to mice fed a chow diet (Fig. 2a, c), plasma and liver FA profiles were not significantly different between GF and SPF mice fed the experimental control diet containing cellulose (Supplementary Figure 3A, B). These results confirm that the observed lipid metabolic differences, particularly the generation of MUFA, in SPF mice depends on a degradable fiber source. In agreement, SPF mice fed with control diet containing 14% of fiber had higher levels of MUFA, but lower contents of PUFA compared to SPF mice fed a standard chow diet with 5% fiber (Supplementary Figure 3C, D).

In summary, these data provide strong evidence that the gut microbiota promotes hepatic FA metabolism by providing a substantial amount of FA 2:0 as precursor for synthesis of C16 and C18 FA.

## Discussion

This multi-omics analysis of hepatic and gut samples from mice was performed to determine the effect of gut microbiota on host lipid metabolism. We profiled GF and SPF mice, and the key findings were verified and further analyzed in antibiotic-treated mice. To the best of our knowledge, this is the first study to report the comprehensive proteome, phosphoproteome, and quantitative lipidome in the liver of GF mice.

We observed that the presence of microbes increases desaturation of palmitate (FA 16:0 n-7) by SCD1 and elongation of γ-linoleic acid (FA 18:3 n-6) to dihomo-γ-linolenic acid (FA 20:3 n-6) by ELOVL5. Interestingly, SCD1 activity has been linked to various diseases including diabetes, hypertriglyceridemia, cardiovascular disease, steatosis, bone health, and cancer[29]. ELOVL5 is considered to be a crucial cellular control point for PUFA synthesis[30], and corresponding knockout mice were shown to develop fatty liver due to elevated hepatic triglyceride generation[31]. FASN protein, phosphoprotein, and mRNA abundances were higher in hepatic samples of colonized animals, which clearly support findings from the Bäckhed lab showing that colonized/conventional animals have elevated hepatic *Fasn* mRNA expression that leads to increased triacylglyceride synthesis[14,15]. Nuclear translocation of SREBP1C controlling FASN, ELOVL5, and SCD1 activity is inhibited by a complex of PPARA and PPARGC1A with LPIN1 and 2[19–21], which had lower expression in SPF mice in our experiments (Fig. 3c). Hepatic ELOVL5 activity is repressed by dietary PUFA, but induced by dietary oleic acid[32,33]. Thus, it can be speculated that MUFA generated by SCD1 together with decreased fractions of PUFA lead to increased ELOVL5 activity and enhanced FA 20:3 n-6 levels observed in SPF animals, although this has to be clarified in further studies.

Our data also show that the altered FA metabolism is reflected in the glycerophospholipid profiles, as the precursor/product ratios of mono-unsaturated/saturated FA species strongly correlate with mono-unsaturated/saturated PC species (Fig. 3e). Using

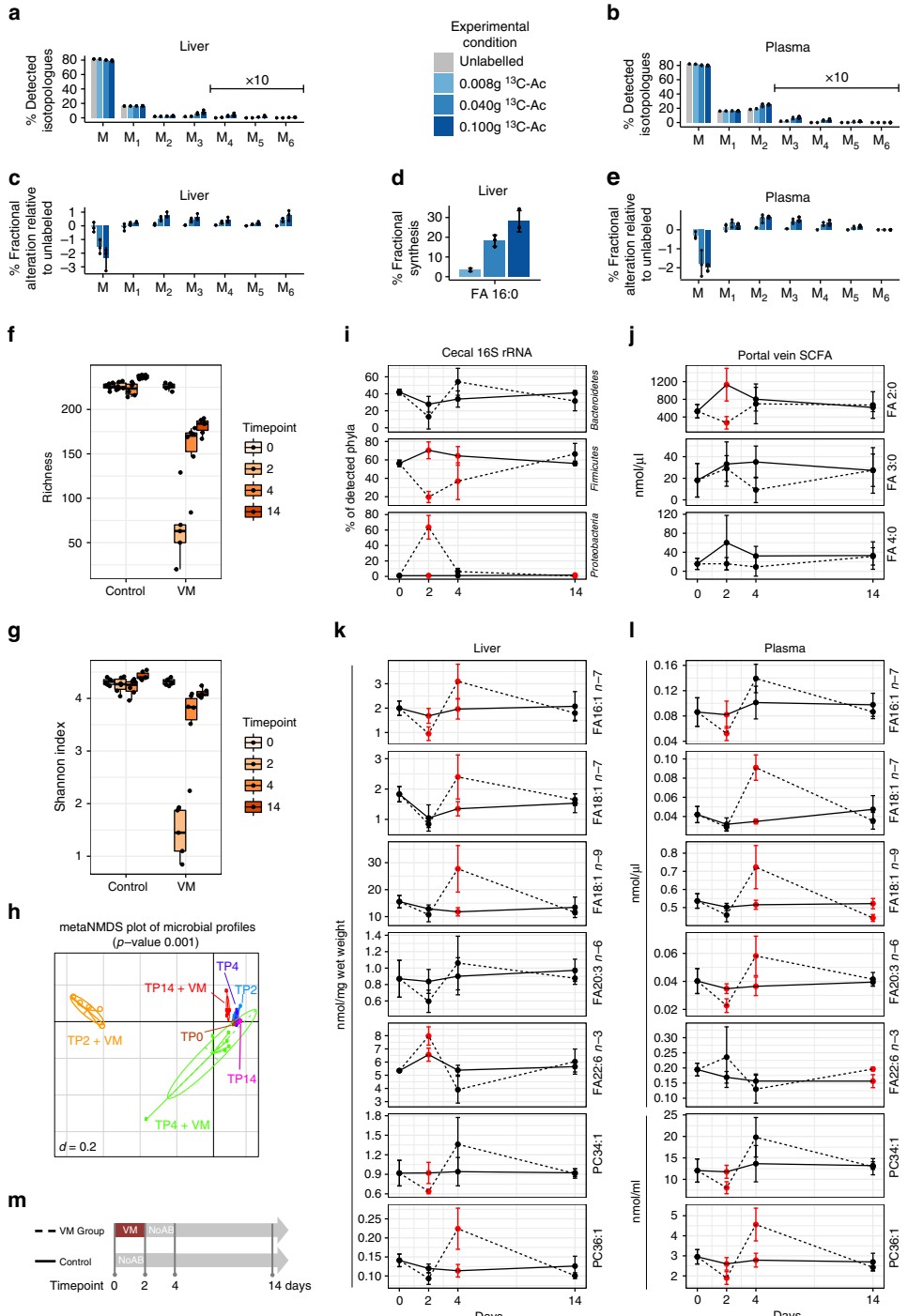

**Fig. 5** Fiber-derived FA 2:0 is precursor for hepatic synthesis of fatty acids and glycerophospholipids. SPF mice were supplemented with $^{13}$C-FA 2:0 via oral gavage with the indicated amount ($n = 3$/group), samples were taken 4 h after gavage. **a** Isotopologue distribution of FA 16:0 in liver and **b** plasma. **c** Fractional alterations (%) of $M_0$, $M_1$–$M_6$ isotopologues relative to unlabeled control in liver and **e** plasma. **d** Fraction of de novo synthesized FA 16:0 in liver. SPF mice were treated for 2 days with VM (TP2), additional 2 (TP4) or 10 days (TP14) without antibiotics ($n = 6$/TP); control SPF mice did not receive antibiotics ($n = 6$/TP). **f** Alpha-diversity analysis shown as richness counts and **g** Shannon effective counts. **h** Multidimensional scaling showing differences in the phylogenetic makeup of microbiota between samples (β-diversity) based on general UniFrac distances. **i** Composition of major phylas determined in cecum content. **j** Portal vein SCFA levels. **k** Levels of selected MUFA, PUFA and MUPC in liver and **l** plasma. AB: antibiotics, Ac: acetate, VM: vancomycin in combination with metronidazole. Red dots and error bars in **i**–**l** indicate a significant difference ($p < 0.05$) between the control and VM group. **m** Experimental setup for the antibiotics experiment. In boxplots the thick lines represent the medians, the upper and lower lines of the boxes show the 25% and 75% quartiles and the whiskers are 1.5 times the interquartile range of the data. In barplots the error bars show the standard deviations. The dot plots in **i**–**l** show the mean and the standard deviation per group and condition

mass spectrometric profiling of different tissues and blood collected from GF and conventionally raised mice, a previous study reported that germfree animals have altered TAG and phospholipid profiles[34]. A part of our results is supported by these findings (e.g., PC 34:1). However, the systematic shift from saturated or polyunsaturated to mono-unsaturated lipid species promoted by gut microbiota as identified in this study was not observed. In our datasets, the alterations in FA 20:3 *n*-6 match LPC 20:3 and CE 20:3 (higher in colonized mice), and changes in FA 22:6 *n*-3 (higher in germfree animals) match CE 22:6 and LPC 22:6. FA 22:6 *n*-3 is of particular interest, since it is the precursor for specialized pro-resolving mediators that are potent anti-inflammatory molecules[35]. Phospholipids enriched in FA 22:6 *n*-3 facilitate membrane deformation, which is essential for membrane fission and supports rapid endocytosis[6,7]. In contrast to lipid species profiles, the concentration of lipid classes, including PC, was similar in colonized and un-colonized animals, suggesting that biosynthesis of glycerophospholipids per se is not influenced by gut microbiota. This hypothesis needs to be evaluated in further studies using dynamic stable isotope experiments to profile FA and phospholipid syntheses[36].

To investigate the effect of antibiotics on FA metabolism, we compared the composite classification score with the gene expression data for *Scd1* and *Elovl5*. We found that, in general, antibiotics have a major effect on hepatic FA metabolism, although the treatments can be ranked according to their effect from highest to lowest: VM > M > V > A. We could not identify any direct associations between the classification score and single OTUs. However, it is known that during antibiotic treatment the total bacterial load in the gut is severely reduced, particularly during the first days of treatment, but recovers after one to five weeks in mice and humans[37,38]. Because we chose a short-term treatment of two days, we conclude that the level of hepatic FA desaturation by SCD1 and elongation by ELOVL5 might depend on the gut microbial load. This assumption is strongly supported by the recent finding that toll like receptor 5 (TLR5) deficient mice, which exhibit a ~3-fold higher bacterial load and are prone to develop microbiota-dependent metabolic syndrome have increased hepatic SCD1 expression and oleic acid (FA18:1 *n*-9) levels[39].

Application of an in vivo stable isotope labeling strategy in combination with mass spectrometric analysis revealed that FA 2:0 originating from the gut lumen is used as precursor for hepatic synthesis of FA, which are released into circulation, also as complex lipids. This data are supported by significantly elevated serum lipid levels found in human subjects after rectal infusion of SCFA[40]. $^{13}$C-FA 2:0 doses applied in the present study are within the physiological range of colonic SCFA levels typically found in mice (0.02 g per oral gavage relates to a concentration of ~200 mM in the gastrointestinal tract)[22,41].

Modulation of *Bacteroidetes* and *Firmicutes* contents and potentially the total gut bacterial load (discussed before) by a short-term antibiotic treatment demonstrated a correlation of FA 2:0 levels to liver MUFA and the PUFA fraction. We chose VM, since this antibiotic combination had most pronounced effects on the composite classification score, SCD1 expression, MUFA and *Bacteroidetes* (Fig. 4). For the production of SCFA, it is important that the gut microbiota work as a community[22,42]. Thus, we conclude that alterations in gut microbial diversity significantly contribute to SCFA levels in our study. At TP2 + VM specifically the fraction of *Proteobacteria* massively rose (Fig. 5i). Previous studies suggested that *Proteobacteria* can be associated with FA 3:0 (but not FA 2:0) fitting very well to the observed changes in portal vein FA 3:0 levels (Fig. 5j)[43,44]. At TP4, MUFA and MUPC fractions reached even higher concentrations than at TP0 (Fig. 5k, l) suggesting a massively induced FA synthesis after antibiotics displacement to restore the initial FA and PC profile.

When mice were fed with a diet containing purified cellulose as fiber source, the FA profiles of GF and SPF mice were similar (Supplementary Figure 3A, B). Purified cellulose is rarely degradable, fecal and portal vein concentrations of SCFA from mice fed cellulose are ~50% lower than in mice fed a degradable fiber source[22,28]. When SPF mice were fed a diet containing elevated fractions of degradable fiber, gut microbiota-mediated effects were boosted compared to SPF mice fed a regular diet confirming the importance of a degradable fiber source for gut microbiota driven FA synthesis in the liver. It was reported previously that colonic formation of SCFA depends on the amount of dietary fiber ingested[22,45].

In summary, we could show that microbial colonization promotes hepatic FA metabolism, which is also reflected in changes in glycerophospholipid species profiles. An important precursor for hepatic long-chain FA is FA 2:0 generated from degradable dietary fiber sources by the gut microbiota. These colonization-driven alterations in host lipid metabolism might not only influence biophysical membrane properties but also signaling processes such as eicosanoid generation and thus affect general physiology as well as inflammatory and metabolic diseases.

## Methods

**Mouse housing**. Studies were performed in SPF and GF C57BL/6 N mice (female) housed at 22 ± 1 °C and 50–60% relative humidity with a 12 h light–dark cycle . Mice were fed a chow diet (autoclaved, V1534, Ssniff) ad libitum and killed at 10 weeks of age. SPF mice were kept in individually ventilated cages, and GF mice were housed in open cages within flexible film isolators ventilated via HEPA-filtered air at the ZIEL Institute for Food & Health. Oligo-MM[12] and GF C57BL/6 J mice (male) were housed in open cages within separate rigid isolators ventilated via HEPA-filtered air at the Helmholtz Centre for Infection Research. Mice were exposed to a 12 h light–dark cycle, fed a chow diet (irradiated, V1124-927, Ssniff) ad libitum and killed at 8 weeks of age. Sterility of GF mice was routinely confirmed by culturing and microscopic observation of feces after Gram-staining. In addition, 16 S rRNA gene-targeted PCR of GF cecal content was performed at the end of the study.

All mouse experiments were performed according to the relevant ethical guidelines. The breeding and experimental use of mice in the facilities at the Technische Universität München (School of Life Sciences Weihenstephan) was approved by the local institution in charge (Regierung von Oberbayern; approval number 55.2-1-54-2531-99-13 and 55.2-1-54-2532-17-2015; 55.2-1-54-2532-192-2016). For all mouse experiments animal group/sample sizes were estimated according to personal experience; the group allocation of the animals was not blinded (practically impossible), but mice were randomly chosen for the different antibiotic treatments and dietary interventions.

**Antibiotic treatment**. SPF C57BL/6 N mice were fed a chow mash containing ampicillin (1 g/L), vancomycin (1 g/L), metronidazol (1 g/L), or a combination of vancomycin (0.25 g/L) and metronidazol (1 g/L) ad libitum for two days at the age of eight weeks (Sigma Aldrich, Fluka) without any additional food (per group: male/female, 1/1).

**In vivo stable isotope labeling experiments**. SPF C57BL/6 N mice (female; chow diet, 11 weeks of age) were fasted for 2 h before application of 0.008 g, 0.040 g and 0.100 g 1-$^{13}$C-acetate dissolved in 0.2 ml NaCl solution (0.9%) via oral gavage. Mice were killed 4 h after oral gavage.

**Dietary intervention experiments**. To study the effects of non-degradable dietary fiber, SPF and GF C57BL/6 N mice (per group: male/female, 1/1) were fed at the age of 6 weeks a purified control diet with comparable carbohydrate, fat and protein content as chow but with purified cellulose (5%) as fiber source (autoclaved, S5745-E702, Ssniff) for 2 weeks. To study the effects of an altered dietary fiber content, SPF C57BL/6N mice were fed at the age of 6 weeks either a standard chow diet (5% grain-soybean-based crude fiber extract; V1534, Ssniff) or control diet comparable to chow, but with elevated content of degradable fiber (14% grain-soybean-based crude fiber extract; V1574, Ssniff) for 2 weeks.

**RNA isolation and quantitative RT-PCR analysis**. Total RNA was extracted from total tissue using the RNEasy Mini Kit (Qiagen). The purity and integrity of the RNA were assessed using the Agilent 2100 bioanalyzer (Agilent Technologies). For real-time PCR, 2 µg RNA was transcribed into cDNA using the Reverse Transcription System from Promega. Real-time quantitative RT-PCR analysis was performed using the Light Cycler LC 480 (Roche). GAPDH was used as a reference gene. The following primers were used: GAPDH (for: 5′-TGCCTCTTCGGGA

TTTTCTACTAC-3′; rev: 5′-TGGAACGCCATGGTGTTGGC-3′), ELOVL5 (for: 5′-ATGTTCTATGAGTTGGTGACAGGT-3′; rev: 5′-GTAGTACCACCAGAG GACGC-3′), FASN (for: 5′-ATGAAGCTGGGCATGCTCAG-3′; rev: 5′-CCGGC ATTCAGAATCGTGGC-3′), SCD1 (for: 5′-TGCCTC TTCGGGATTTTCTACT AC-3′; rev: 5′-TGGAACGCCATGGTGTTGGC-3′). Relative quantification was carried out using the LightCycler 480 SW 1.5.1 (Roche).

**Transcriptomics.** Total RNA (300 ng) was processed using the One-Color Quick Amp Labeling Kit according to the manufacturer's instructions (Agilent). cRNA quantity and labeling efficiency was checked using Nanodrop (PeqLab). The Whole Mouse Genome Microarray 4x44K Kits (G4122F, Agilent) was used for hybridi-zation. Scanning of arrays was performed using the G2565CA (Agilent) (5 μM, single pass, 20 bit, no XDR). Microarray scan data were extracted using the Feature Extraction software 10.7.3.1 (Agilent).

**Proteomics and phospho-proteomics.** Liver samples containing 300 μg of protein were digested with trypsin overnight. After digestion, the samples were acidified with TFA to pH 2 and desalted according to the manufacturer's instructions using a SepPack column [C18 cartridges, Sep-Pak Vac, 1 cc (50 mg)] (Waters Corp., Eschborn, Germany). Eluates were dried down and stored at −80 °C.

TMT labeling was performed according to the manufacturer's instructions (TMT10plex™ Isobaric Label Reagent Set, Thermo Scientific). Briefly, samples were resolved in 40 μL of 50-mM TEAB solution (1.0 M, pH 8.5) (Sigma); 60 μL TMT stock solution was added to 300 μg protein digest for each sample.

Before phosphopeptide enrichment, the 10 TMT-labeled samples were combined. Phosphopeptide enrichment was performed using an analytical Fe-IMAC column (4 x 50 mm ProPac IMAC-10, Thermo Fisher Scientific) connected to an HPLC system (AEKTA Explorer FPLC system, Amersham Biosciences Pharmacia)[46,47]. The phosphopeptide fraction and the flow-through were collected according to the UV signal (280 nm), dried down, and stored at −80 °C.

Reversed phase fractionation was performed only for the phospho-peptide-enriched samples[48]. Sample fractionation was performed by sequential elution of the bound peptides using each of the six buffers (40 μL) containing an increasing concentration of ACN (5%, 7.5%, 10%, 12.5%, 15%, 17.5%, and 50% ACN in 25 mM NH$_4$FA). The desalted sample flow-through fraction was combined with the 17.5% ACN fraction and the 5% ACN fraction combined with the 50% ACN fraction. Samples were dried down and were stored at −20 °C before LC-MS/MS measurement.

Only the flow-through of the Fe-IMAC column phosphopeptide enrichment was used for further separation using the hydrophilic strong anion exchange separation (hSAX)[49]. A Dionex Ultimate 3000 HPLC system (Dionex Corp., Idstein, Germany) equipped with an IonPac AG24 guard column (2 × 50 mm, Thermo Fisher Scientific) and a IonPac AS24 strong anion exchange column (2 x 250 mm, Thermo Fisher Scientific) was used for the hSAX separation of the full proteome sample (IMAC flow-through). A total of 36 hSAX fractions (1 min per fraction) was collected manually, pooled according to the UV trace, and desalted, giving a total of 24 fractions for LC-MS/MS measurement[50].

The 24 hSAX fractions (50 μL of 0.1% FA) were analyzed by LC-MS/MS using a Thermo Ultimate 3000 HPLC (Thermo Scientific, Germering, Germany) coupled to a Q-Exactive HF instrument (Thermo Scientific, Bremen, Germany). Each sample (5 μL) was delivered to the trap column (100 μm ID × 2 cm, 5 μm C18 resin; Reprosil Pur AQ, Dr. Maisch) at a flow rate of 5 μL/min in solution A (0.1% FA in water) for 10 min. For peptide separation, peptides were transferred to the analytical column (75 μm × 40 cm, 3 μm C18 resin; Reprosil, Pur AQ Dr. Maisch) and separated at a flow rate of 300 nL/min using a 110 min gradient (2–4% solution B in 1 min; 4–32% in 102 min). The Q-Exactive HF was operated in the data-dependent mode, automatically switching between MS1 and MS2. The full-scan MS spectra from 360 to 1300 m/z was acquired at 60,000 resolution with an automatic gain control (AGC) target value of $3 × 10^6$ charges and a maximum injection time of 50 ms for MS1. Up to 25 precursor ions were allowed for fragmentation in tandem mass spectra using a normalized collision energy of 33. MS2 spectra were acquired at 30,000 resolution, with an AGC target value of $2 × 10^5$ charges and max injection time of 57 ms. The precursor ion isolation width was set to 1.0 Th and the dynamic exclusion set to 20 s.

The 6 IMAC fractions were reconstituted in 10 μL of 1% FA in 50 mM citrate. LC-MS/MS measurements were performed using an Eksigent NanoLC-Ultra 1D + coupled to a Q-Exactive Plus instrument (Thermo Scientific, Bremen, Germany). A 5-μL sample of IMAC-enriched phosphopeptides was delivered to the trap column (100 μm ID × 2 cm; 5 μm C18 resin; Reprosil Pur AQ, Dr. Maisch) at a flow rate of 5 μL/min in solution A (0.1% FA in water) for 10 min. Peptides were transferred to the analytical column (75 μm × x 40 cm; 3 μm C18 resin; Reprosil, Pur AQ Dr. Maisch) and separated at a flow rate of 300 nL/min using a 110-min gradient (2% to 4% solution B in 2 min; 4–32% in 102 min) for peptide separation. The Q-Exactive Plus was operated in data-dependent mode, automatically switching between MS1 and MS2. The full-scan MS spectra from 360 to 1300 m/z was acquired at 70 000 resolution with an AGC target value of $3 × 10^6$ charges and a maximum injection time of 100 ms for MS1. Up to 20 precursor ions were allowed for fragmentation in tandem mass spectra. MS2 spectra were acquired at 35,000 resolution and a normalized collision energy of 33. An AGC target value of $2 × 10^5$

charges and a maximum injection time of 100 ms were used. We set the precursor isolation window to 1.0 Th and the dynamic exclusion to 20 s.

Peptides and proteins were identified by comparing the raw data to the UniProtKB mouse database, version v06.06.14 (35098 sequences) using MaxQuant version 1.5.2.8 and its built-in Andromeda search engine for peak detection and quantification[51]. Search parameters for the full proteome samples were as follows: full tryptic specificity, up to two missed cleavage sites; carbamidomethylation of cysteine residues was set as a fixed modification; N-terminal protein acetylation and methionine oxidation were set as variable modifications. TMT10plex was used for quantification. Mass spectra were recalibrated within MaxQuant (first search 20 ppm precursor tolerance) and subsequently searched again with a mass tolerance of 6 ppm; fragment ion mass tolerance was set to 20 ppm. Search results were filtered to a maximum FDR of 0.01 for proteins and peptides. A peptide length of at least seven amino acids was required. Search parameters for phosphopeptide enrichment PAL data were similar to the full proteome data, with the phosphorylation of serine, threonine, and tyrosine residues as additional variable modifications.

**Total fatty acid analysis.** Fatty acid methyl esters (FAMEs) were generated by acetyl chloride and methanol treatment and extracted with hexan[52]. Total FA analysis was carried out using a Shimadzu 2010 GC-MS system. FAMEs were separated on a BPX70 column (10-m length, 0.10-mm diameter, 0.20-μm film thickness) from SGE using helium as the carrier gas. The initial oven temperature was 50 °C and was programed to increase at 40 °C/min to 155 °C, 6 °C/min to 210 ° C, and finally 15 °C/min to 250 °C. The FA species and their positional and cis/ trans isomers were characterized in scan mode and quantified by single ion monitoring, to detect specific fragments of saturated and unsaturated FAs (satu-rated, m/z 74; mono-unsaturated, m/z 55; di-unsaturated, m/z 67; poly-unsatu-rated, m/z 79). The internal standard was non-naturally-occurring C21:0 iso. For all lipidomic experiments, samples belonging to same experiment were analyzed in one batch under standardized conditions; the group allocation was blinded during sample preparation and peak integration.

Enrichment of 1-$^{13}$C-FA 2:0 in FA 16:0 was analyzed by mass isotopomer distribution analysis (MIDA) using single ion monitoring of molecular ions (M: m/ z 270; $M_1$-$M_8$: m/z 271-278). Fractional syntheses representing the fraction of newly synthesized FA 16:0 after 4 h were calculated by MIDA using the excess mass ratio of the isotopologues $M_3/M_2$ relative to the unlabeled control[36,53].

**Short chain fatty acid analysis.** SCFA were quantified by LC-MS/MS after derivatization to their 3-nitrophenylhydrazones as described by Han et al.[54]. In brief, 10 μl of an internal standard mixture containing 20 μg/ml each $D_4$-acetic acid, $D_5$-propionic acid, $D_7$-butyric acid (Cambridge Isotope Laboratories, USA) were added to 10 μl of portal vein plasma and mixed. 100 μl acetonitrile were added and centrifuged after mixing. 50 μl of the supernatant were derivatized for 30 min at 40 °C with each 20 μl of 200 mM 3-nitrophenylhydrazine hydrochloride and 120 mM N-(3-dimethylaminopropyl)-N′-ethylcarbodiimide hydrochloride. The reac-tion was quenched by addition of 200 μl 0.1 % formic acid. LC separation was performed using a Kinetex® 2.6 μm XB-C18, 50×2.1 mm (Phenomenex, Torrance, CA, US) with water as mobile phase A and acetonitrile as mobile phase B both containing 0.1% formic acid. Gradient elution started with 90% A with a linear increase to 81% A at 0.3 min follow by an increase to 78%A at 2.5 min. Column was cleaned with 100% B from 2.6–3.0 min and re-equilibrated from 3.1 to 4 min with 90% A. The column flow was 500 μl at 60 °C and 10 μl sample were injected. The method included acetic acid, propionic acid, butyric acid and iso-butyric acid. Butyric acid and iso-butyric acid were separated by LC. For quantification a cali-bration lines were generated by addition of SCFA to human plasma.

**Glycerophospholipid and cholesterol analysis.** Lipids were extracted according to the procedure described by Bligh and Dyer in the presence of non-naturally-occurring lipid species as internal standards. The chloroform phase was dried in a vacuum centrifuge and dissolved as described below for quantitative lipid analysis.

Lipids were quantified by electrospray ionization tandem mass spectrometry (ESI-MS/MS) in positive ion mode[55]. In brief, samples were analyzed by direct flow injection using a HTS PAL autosampler, an Agilent 1100 binary pump, and a triple quadrupole mass spectrometer (Quattro Ultima, Micromass). A precursor ion scan of m/z 184 specific for phosphocholine-containing lipids was used for PC, sphingomyelin (SM), and lysophosphatidylcholine (LPC)[56]. The following neutral losses were applied: Phosphatidylethanolamine (PE), 141; phosphatidylserine (PS), 185; phosphatidylglycerol (PG), 189; and PI, 277. PE-based plasmalogens (PE P) were analyzed according to the principles described by Zemski Berry et al.[57]. Sphingosine-based ceramides (Cer) were analyzed using a fragment ion of m/z 264. FC and CE were quantified using a fragment ion of m/z 369 after selective derivatization of FC using acetyl chloride. Corrections for isotopic overlap of lipid species and data analysis using Excel Macros were performed for all lipid classes. Non-naturally occurring lipid species were used as internal standards in all analyses. Quantification was performed by the addition of a standard to cell homogenates, using a number of naturally-occurring lipid species as standards for each lipid class. Lipid species were annotated according to the recently published proposal for shorthand notation of lipid structures derived from MS[58]. Glycerophospholipid species annotation was based on the assumption of even-

numbered carbon chains only. SM species annotation is based on the assumption that a sphingoid base with two hydroxyl groups is present.

**Analysis of sphingolipids and minor glycerophospholipids.** Lipids were extracted using butanol in the presence of non-naturally occurring internal standards[59,60]. In brief, lipid extracts were subjected to hydrophilic interaction chromatography (HILIC) coupled to MS/MS (HILIC-MS/MS) to quantify hexosylceramides (HexCer), lactosylceramides (LacCer), sphingoid bases, sphingosylphosphorylcholine (SPC), cardiolipin (CL), bis(monoacylglycero)phosphate (BMP), phosphatidylglycerol (PG), and phosphatidic acid (PA). For each lipid class, non-naturally occurring internal standards were added, and quantification was achieved by calibration lines generated by the addition of naturally occurring lipid species to the sample matrix. Deisotoping and data analysis for all lipid classes were performed using self-programed Excel Macros[60].

**High-throughput 16S rRNA gene amplicon analysis.** Metagenomic DNA was obtained from cecal content after mechanical lysis followed by purification according to a published protocol[61] modified as follows: cecal content in 600 μL of stool DNA stabilizer (Stratec Biomedical AG) was transferred to a 2-mL screw-cap tube containing 500 mg zirconia/silica beads (0.1 mm; BioSpec Products), 250 μL 4 M Guanidinethiocyanate (Sigma-Aldrich, Germany), and 500 μL 5% N-laurolylsarcosine (Sigma-Aldrich, Germany). Samples were mixed, incubated for 60 min at 70 °C with constant shaking, and bacterial cells were disrupted by mechanical lysis using a FastPrep-24 (3 times, 40 sec, 6.5 m/sec) (MP Biomedicals) fitted with a cooling adapter. After addition of 15 mg Polyvinylpolypyrrolidone (PVPP, Sigma-Aldrich), the suspension was vortexed and centrifuged (3 min; 15.000 × g; 4 °C). The supernatant (500 μL) was transferred to a new Eppendorf tube, mixed with 5 μL RNase (stock concentration, 10 mg/ mL; VWR International) and incubated for 20 min at 37 °C with constant shaking. Genomic DNA was purified using NucleoSpin® gDNA columns (Macherey Nagel GmbH & Co. KG, Germany) following the manufacturer's instructions. DNA quantity and quality was measured using a NanoDrop® instrument (Thermo Fisher Scientific Inc., Germany).

Libraries were constructed in a semi-automated manner using a Biomek-4000 pipetting robot (Beckmann Coulter Biomedical GmbH). The V3/V4 region of the 16 S rRNA genes was amplified (25 cycles) from 24 ng of metagenomic DNA using primer 341 F and 785 R in a 2-step procedure to limit amplification bias[62,63]. Libraries were double-barcoded (8-nt index on both the forward and reverse 2nd-step primer)[64,65]. Amplicons were purified using the AMPure XP system (Beckmann Coulter Biomedical GmbH), pooled in equimolar amounts with 25% (v/v) PhiX library and sequenced in paired-end modus (PE275) using a MiSeq system (Illumina).

Raw sequence reads were processed using IMNGS (www.imngs.org)[66,67], a pipeline developed in-house based on UPARSE[68]. Parameters were as follows: barcode mismatches, 2; expected error, 3; Phred quality threshold, ≥3; trimming score, 3; trimming length, 10 nt; minimum sequence length, 300 nt; maximum sequence length, 600 nt (see IMNGS website for further information). OTUs were clustered at 97 % sequence similarity; only those occurring at a relative abundance ≥0.25 % total reads in at least one sample were further analyzed. For each OTU, the final taxonomy was assigned using the most detailed classification in SILVA[69] and RDP[70]. Downstream statistical analysis of diversity and composition data were performed using Rhea[71]. EzTaxon was used to identify OTUs with significant differences in relative abundance between feeding groups[72].

**Statistical analyses of transcriptomics.** The raw data were analyzed in R (version 3.3.1 R Core Team (2016). R: A language and environment for statistical computing. R Foundation for Statistical Computing, Vienna, Austria. URL https://www.R-project.org/) using limma[73] as suggested in the limma manual. Briefly, the image files were read into R, and the raw values were background-corrected using the "normexp" method with an offset of 50. The raw data were quantile normalized between the arrays, and the average of replicate spots per gene was calculated. A linear regression model was calculated using the design of the experiment. P-values, log fold-changes and Benjamini–Hochberg-corrected p-values were calculated using empirical Bayes moderated t-statistics based on the previously-calculated linear fit as implemented in limma, a standard package to analyze transcriptomics data. A gene was said to be significant when the Benjamini–Hochberg-corrected p-value was below 0.05. No samples were excluded. Mean and standard deviations were shown to indicate the variation within each analyzed group.

**Statistical analyses of proteomics and phosphoproteomics.** Proteomics and phosphoproteomics data were analyzed in the same way but performed separately. Briefly, potential contaminants and reverse sequence peptides were removed from the data, and the values were normalized for the total peptide intensity measured per sample. Missing values were replaced by the minimum intensity value per analyzed peptide, and all values were log₂ transformed to ensure that the values were normally distributed. The p-values for differential peptide intensities were calculated using a standard two-sided unpaired t-test assuming unequal variances, variances, since there were no variables for which the analyses had to be adjusted. The p-values were adjusted for multiple testing using the Benjamini–Hochberg

method to adjust for the FDR. Peptides were said to be significantly differentially expressed if the Benjamini–Hochberg adjusted p-value was <0.05. No samples were excluded. Mean and standard deviations were shown to indicate the variation within each analyzed group.

**Integration of transcriptome, proteome, and phosphoproteome.** The omics datasets were merged based on the gene symbol (e.g., FASN). If multiple peptides or transcripts mapped to the same gene, the transcript or peptide with the highest interquartile range was chosen. Uniprot ID conversion was performed using the uniprot conversion tool (http://www.uniprot.org/uploadlists/) with the UniProtKB AC/ID that returns mapped IDs and gene names for the peptides.

**Statistical analyses of lipidomic data.** The lipids per class and per matrix were analyzed separately (i.e., FAs in liver were analyzed separately from all other lipid classes and organs). The lipid classes were quotient normalized and the sums of species, e.g., polyunsaturated fatty acids, were calculated. All data were then log₂ transformed to ensure that they were normally distributed. A Fisher test was used to assess whether missing values were missing at random or if there was an underlying bias. Lipid species were excluded if they were undetectable in less than 50% of the samples per group, samples were excluded when they were obvious outliers on a PCA plot or due to obvious technical problems. A standard two-sided, unpaired t-test assuming unequal variances was used to test for significantly different abundances in the conditions. Mean and standard deviations were shown to indicate the variation within each analyzed group.

**Pathway enrichment analysis.** The pathway enrichment analysis for GO and KEGG was performed using the safe package[74]. This method is based on a resampling approach designed to counter the high false-positive rate observed in other methods[75]. Briefly, the expression data, phenotype, and list of functional annotations were provided. Functional enrichments were calculated using the standard options in safe using the "bootstrap.t" method, such that data were bootstrapped 200 times to ensure that the results were not inflated by false positives. The Benjamini–Hochberg method to calculate the false discovery rate (FDR) was used to account for multiple testing (significance, $p_{adj} < 0.05$). Safe analyses returned a list of pathways or functional annotations that were affected by differentially expressed genes.

**Classification score calculation.** To identify the smallest number of FAs indicating germfree status, we used penalized logistic regression. For each sample i, we coded the response variable as $y_i \in \{0, 1\}$ where 0 was the germfree status. We used the combined Experiment 1 and 2 (GF: n = 18; SPF: n = 20) as a training dataset and combined Experiment 3 and the Antibiotics Experiment as a test dataset. The conditions in the antibiotics dataset were modeled as germfree when the samples were treated with antibiotics, and the controls were modeled as specific pathogen-free. FAs and the experiment name were modeled as independent variables.

$$y_i \sim \beta_{\text{Experiment}} * \text{Experiment}_i + \beta_{\text{FA}_1} * \text{FA}_1 \\ + \beta_{\text{FA}_2} * \text{FA}_2 + \ldots + \beta_{\text{FA}_n} * \text{FA}_n \tag{1}$$

The experimental variable was a fixed covariate and was not penalized. The penalty factor λ, used to reduce the number of covariates in the model, was calculated by a cross-fold validation using the leave-one-out approach from which the lambda.se1 was extracted. The covariates remaining after penalization were used to classify the conditions in the test dataset. The models were calculated in R (version 3.3.1) using the glmnet package (version 2.0-5)[76].

**Correlations.** All correlations were calculated in R using cor() with options method = "spearman" or "pearson" for the Spearman's or Pearson's correlation coefficient, respectively. Pearson's correlation is a measure of a linear correlation in the data, while Spearman's correlation coefficient is based on the ranked values rather than the values themselves.

**Code availability.** Computer code can be made available from the corresponding authors on reasonable request. All statistical analyses were performed in R version 3.3.1. Package versions were stated where applicable.

**Data availability.** The transcriptomic, proteomic and phosphoproteomic data are included in this article as Supplementary Information (Supplementary Data 1-3). Lipidomic datasets analyzed in this study are included in this article as tables, main and Supplementary Figures, and Supplementary Data. Microarray data are also deposited at ArrayExpress under accession code E-MTAB-6079. The mass spectrometry proteomics data have been deposited to the ProteomeXchange Consortium via the PRIDE partner repository with the dataset identifier PXD010412. All remaining data will be made available by the corresponding authors upon reasonable request.

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

## Acknowledgements

This work was supported by a Deutsche Forschungsgemeinschaft (DFG) grant for a temporary position as principal investigator (J.E., EC 453/1-1) and the DFG priority program "SPP 1656- Intestinal microbiota (EC 453/2-1, LI 923/4-1, STR 1343/2)"; grants from the European Union's Seventh Framework Programme under grant agreement numbers 305280 (MIMOmics) [FP7-Health-F5-2012], 613979 (MyNewGut) and by the Helmholtz Cross-Program Initiative Personalized Medicine "iMED." We thank Ronny Scheundel, Doreen Mueller, Simone Düchtel, Sebastian Roth, Alexander Wolf, Malwine Solecki, Achim Gronow, Caroline Ziegler, and Angela Sachsenhauser for excellent technical assistance.

## Author contributions

J.E. designed the research; J.K. supervised statistics and bioinformatics; J.E. and J.K. acquired funding; J.E., J.K., A.K., and G.L. wrote the paper; A.K. performed statistics and bioinformatics analyses; G.L., J.E., A.G., and S.K. conducted lipidomic analyses; and T.C., Di.H., G.H., H.Y., A.S., Do.H., B.K., D.K., C.S., R.M., T.S., S.S., and H.D. performed experiments and/or contributed materials.

## Additional information

**Competing interests:** The authors declare no competing interests.

