## [Peer Review File · Nature Communications]

Reviewers' comments:

Reviewer #1 (Remarks to the Author):

The authors applied a comprehensive multi-omics analysis of germ-free (GF) and SPF C57BL/6N mice, focusing on liver and plasma samples. Specific antibiotics were also tested in separate experiment in SPF mice, and the effect on hepatic lipid metabolism was examined.

The cross-sectional analysis suggested, based on both proteomic/transcriptomic as well as lipidomic data, that gut microbiota induces FA elongation via ELOVL5 and desaturation via SCD1, leading in particular to changes in phospholipid profiles. The SPF-like lipid signature became more GF-like after the administration of specific antibiotics.

The experimental techniques used in the study are of high standard, and as such provide a valuable contribution to the field. The data are very clearly presented.

The experimental setting is straightforward, but with limited information e.g. on the mouse study setting, e.g. (1) was sterility of GF mice regularly checked and how? (2) if non-identical housing conditions (isolators) were used after weaning, could that also be one factor affecting the results?

The major issue in the study is that too much is concluded based on cross-sectional comparison of two groups of mice in a relatively small study. The observation that fatty acid desaturation and elongation are upregulated in the livers of SPF mice is well supported by the data, but the study does not provide an insight into the potential mechanism behind it. The antibiotics experiments do suggest that different treatments all seem to produce the GF-like profile, but it would be important to see the effect on lipid metabolism after conventionalisation or colonisation with specific strains, and perhaps with respect to genetic/pharmacological targeting of the pathway of interest. The authors did attempt to analyse the microbial composition after antibiotic treatments, but the results did not offer any insight into their effect on lipid metabolism.

Reviewer #2 (Remarks to the Author):

In this manuscript, Dr. Kindt and his/her colleague reported that the gut microbiota promotes hepatic fatty acid desaturation and elongation in mice. By using comprehensive multi-omics analysis, this research showed that gut microbiota influences to host hepatic lipid metabolism, therefore, GF mice contained more abundant saturated and poly-unsaturated lipids, whereas SPF mice contained mono-unsaturated lipids.

Multi-omics analysis such as integrating transcriptomic, proteomic, phosphoproteomic, and lipidomic analyses in liver and plasma samples from GF and SPF mice, are first report, and important for understanding of commensal host-bacterial relationship. However, in this study, authors have not been able to identify gut microbiota-related key factors for inducing MUFA generation by SCD-1 and PUFA elongation by ELOVL5.

Gut microbes play an important role in the energy harvest. Does difference of energy conditions between SPF and GF mice, influence to hepatic lipid metabolism? How about influence of gut absorbed-nutrients and gut microbial metabolites such as short-chain fatty acids? Authors also should perform water soluble metabolomics of portal blood in addition to lipids metabolome of plasma.

Moreover, to clarify direct influence of gut microbes to host hepatic lipid metabolism, authors should examine relationship between host hepatic lipid metabolism and change of gut microbial composition and metabolites on dietary difference (for example; low or high fiber, normal diet or high fat diet) but not between colonized and germ free.

I think that by integrating these data, authors can find key factors influenced to host hepatic lipid metabolism in gut microbes and metabolites.

Point-to-point response to the referees (NCOMMS-17-28020)

As some of the points raised by the Reviewers were similar, some answers are replicated. Most work done for the revision of the manuscript was related to Comment 1, which basically answers to most of the reviewer`s questions.

Our new investigations were added and discussed including a new figure (Fig. 5) and two supplementary figures (Fig. SI3 and SI4). All changes made to the original manuscript are marked in red.

We would really like to thank the reviewers for their helpful comments, which substantially improved this manuscript. We hope that we could provide the additional information and clarifications requested.

Editor/Reviewer I and II:

Comment 1:

“In particular, it is our editorial view that the revised manuscript should include new experiments providing additional mechanistic insight, as recommended by the reviewers.”

Now, we provide additional mechanistic insights by showing for the first time that SCFA/acetate (FA 2:0) originating from gut microbial degradation of dietary fiber is a precursor for hepatic synthesis of C16 and C18 fatty acids finally entering the circulation.

Experimental evidence was collected from the following new animal experiments:

- i. To show that acetate produced in the gut lumen is used for hepatic FA synthesis, the following stable isotope labelling strategy was applied: Mice were supplied with different concentrations of ¹³C-labeled acetate via oral gavage→ The enrichment of ¹³C in FA 16:0 of liver and plasma was investigated after mass spec-based analysis.
- ii. To confirm that synthesis of FA/MUFA depends on gut microbial SCFA producers and portal vein FA 2:0 levels, first, *Bacteroidetes* and *Firmicutes* as major SCFA producers were eliminated with antibiotics. Secondly, the antibiotics were displaced to recover the initial conditions/SCFA levels. FA and PC species were analyzed as for previous experiments, portal vein SCFA levels were quantified by LC-MS/MS.
- iii. To demonstrate that the hepatic synthesis of MUFA and PUFA metabolism depend on dietary fiber that is degradable and fermentable by gut microbiota (a) GF and SPF mice were fed a diet containing a non-degradable fiber source/cellulose and (b) SPF mice were fed diets with different contents of degradable fiber.

The following was added to the manuscript:

- **Figure 5** (see last pages of this document)
- **Figure SI4** (see last pages of this document)

- “Mechanistic investigations revealed that acetate originating from gut microbial degradation of dietary fiber serves as precursor for hepatic synthesis of C16 and C18 fatty acids and their related glycerophospholipid species that are also released into the circulation.” (Abstract, p.2)
- “Using in vivo stable isotope labelling experiments and dietary intervention strategies we showed that the short chain fatty acid (SCFA) acetate (FA 2:0) originating from gut microbial degradation of dietary fiber is a precursor for hepatic synthesis of long chain fatty acids and glycerophospholipids containing these fatty acids.” (Introduction, p.4)
- **“The gut microbiota promotes hepatic lipid metabolism by providing acetate as precursor for synthesis of C16 and C18 fatty acids and their related phosphatidylcholine species**

Members of *Bacteroidetes* and *Firmicutes* including *Kineothrix alysoides* (OTU-15) degrade dietary polysaccharides from fiber to short chain fatty acids (SCFA) including acetate (FA 2:0), which reach the liver via the portal vein (den Besten et al., 2013; Haas and Blanchard, 2017). Portal blood FA 2:0 levels are ~ 0.5 mM (FA 2:0/FA 3:0/FA 4:0: ~ 50/3/1) in mice, rats and humans (Cummings et al., 1987; Jakobsdottir et al., 2013).

To ask whether FA 2:0 originating from the gut lumen is a precursor for synthesis of long chain FA in the liver, mice were supplied with different concentrations of ¹³C-labeled FA 2:0 via oral gavage. After 4 h liver and plasma FA 16:0 were analyzed for ¹³C-enrichment using GC-MS. Administration of ¹³C-FA 2:0 clearly enhanced isotopologues M₂-M₆ of FA 16:0 in a dose-dependent way in both liver and plasma (Figure 5A, B) resulting in an increase of the fractional abundances of M₂-M₆ and decrease of monoisotopic M₀ (Figure 5C, E). ¹³C-Acetate also dose-dependently elevated the fraction of newly synthesized FA 16:0 in the liver 4 h after oral gavage from ~4 to 30 % demonstrating a stimulation of the hepatic FA *de novo* synthesis by gut-derived acetate (Figure 5D).

To demonstrate that differences in the lipid profiles of GF and colonized mice (Figure 2, Figure SI3), particularly altered MUFA contents, can be associated with microbial FA 2:0 production in the gut, we next manipulated *Bacteroidetes* and *Firmicutes* as major SCFA producers. SPF mice received a combination of vancomycin and metronidazole (VM; as describe before) for two days (time-point 2, TP2), before they obtained a regular chow diet without antibiotics for additional two (TP4) or 10 days (TP 14). At TP2+VM gut microbial composition and diversity dropped (α - and β -diversity; Figure 5F, G, H). *Bacteroidetes* and *Firmicutes* were almost completely eliminated (Figure 5I) and portal vein FA 2:0 concentrations were reduced 2-fold (Figure 5J). After removal of the antibiotics (TP4 and TP14), the gut microbial ecosystem recovered and baseline FA 2:0 levels (TP0) were reached. Importantly, alterations of liver and plasma MUFA (16:1 *n*-7, FA 18:1 *n*-9, 18:1 *n*-7), MUPC (PC 34:1, PC 36:1) and PUFA (20:3 *n*-6, 22:6 *n*-3) levels clearly followed this trend (Figure 5K, L). These results indicate a direct relation of liver and plasma FA levels to gut microbial SCFA producers and portal blood FA 2:0 levels.

Chow diet used in these experiments comprises polysaccharides from a grain-soybean-based crude fiber extract (5 %) that can be depolymerized and subsequently fermented by gut microbiota to SCFA. To demonstrate that hepatic synthesis of MUFA depends on gut microbial degradation of dietary fiber, GF and SPF mice were fed with an experimental control diet with carbohydrate, fat and protein content comparable to chow, but with 5 % purified cellulose instead of crude fiber. Refined cellulose is practically non-degradable and -fermentable by gut microbiota leading to markedly reduced SCFA levels in portal blood (Flint et al., 2012; Slavin et al., 1981; Weitkunat et al., 2015). In contrast to mice fed a chow diet (Figure 2A, C), plasma and liver FA profiles were not significantly different between GF and SPF mice fed the experimental control diet containing cellulose (Figure SI4A, B). These results confirm that the observed lipid metabolic differences, particularly the generation of MUFA, in SPF mice depends on a degradable fiber source. In agreement, SPF mice fed with control diet containing 14% of fiber had higher levels of MUFA, but lower contents of PUFA compared to SPF mice fed a standard chow diet with 5% fiber (Figure SI4C, D). In summary, these data provide strong evidence that the gut microbiota promotes hepatic FA metabolism by providing a substantial amount of FA 2:0 as precursor for synthesis of C16 and C18 FA.” (Results, p.10-11)

- “Hepatic ELOVL5 activity is repressed by dietary PUFA, but induced by dietary oleic acid (Ducheix et al., 2017; Wang et al., 2005). Thus, it can be speculated that MUFA generated by SCD1 together with decreased fractions of PUFA lead to increased ELOVL5 activity and enhanced FA 20:3 *n*-6 levels observed in SPF animals, although this has to be clarified in further studies.” (Discussion p.12)
- “Application of an *in vivo* stable isotope labelling strategy in combination with mass spectrometric analysis revealed that FA 2:0 originating from the gut lumen is used as precursor for hepatic synthesis of FA, which are released into circulation, also as complex lipids. This data are supported by significantly elevated serum lipid levels found in human subjects after rectal infusion of SCFA (Wolever et al., 1989). ¹³C-FA 2:0 doses applied in the present study are within the physiological range of colonic SCFA levels typically found in mice (0.02 g per oral gavage relates to a concentration of ~ 200 mM in the gastrointestinal tract) (Casteleyn et al., 2010; den Besten et al., 2013).

Modulation of *Bacteroidetes* and *Firmicutes* contents and potentially the total gut bacterial load (discussed before) by a short-term antibiotic treatment demonstrated a correlation of FA 2:0 levels to liver MUFA and the PUFA fraction. We chose VM, since this antibiotic combination had most pronounced effects on the composite classification score, SCD1 expression, MUFA and *Bacteroidetes* contents (Figure 4). For the production of SCFA, it is important that the gut microbiota work as a community (den Besten et al., 2013; el-Khoury et al., 1994). Thus, we conclude that alterations in gut microbial diversity significantly contribute to SCFA levels in our study. At TP2+VM specifically the fraction of *Proteobacteria* massively rose (Figure 5I). Previous studies suggested that *Proteobacteria* can be associated with FA 3:0 (but not FA

2:0) fitting very well to the observed changes in portal vein FA 3:0 levels (Figure 5J) (Islam et al., 2011; Liou et al., 2013). At TP4, MUFA and MUPC fractions reached even higher concentrations than at TP0 (Figure 5K, L) suggesting a massively induced FA synthesis after antibiotics displacement to restore the initial FA and PC profile.

When mice were fed with a diet containing purified cellulose as fiber source, the FA profiles of GF and SPF mice were similar (Figure S4A, B). Purified cellulose is rarely degradable, fecal and portal vein concentrations of SCFA from mice fed cellulose are ~50% lower than in mice fed a degradable fiber source (den Besten et al., 2013; Weitkunat et al., 2015). When SPF mice were fed a diet containing elevated fractions of degradable fiber, gut microbiota-mediated effects were boosted compared to SPF mice fed a regular diet confirming the importance of a degradable fiber source for gut microbiota driven FA synthesis in the liver. It was reported previously that colonic formation of SCFA depends on the amount of dietary fiber ingested (den Besten et al., 2013; Topping and Clifton, 2001).” (Discussion, p.13/14)

- “An important precursor for hepatic long-chain FA is FA 2:0 generated from degradable dietary fiber sources by the gut microbiota.” (Discussion, p.14)

- **“Figure 5:** Enrichment of gut luminal ¹³C-FA 2:0 in hepatic and plasma FA 16:0; association of gut microbial fiber degraders with portal vein SCFA and long chain FA contents of liver and plasma.

SPF mice were supplemented with ¹³C-FA 2:0 via oral gavage with the indicated amount (n=3/group), samples were taken 4h after gavage. (A) Isotopologue distribution of FA 16:0 in liver and (B) plasma. (C) Fractional alterations (%) of M₀, M₁-M₆ isotopologues relative to unlabeled control in liver and (E) plasma. (D) Fraction of *de novo* synthesized FA 16:0 in liver. SPF mice were treated for 2 days with VM (TP2), additional 2 (TP4) or 10 days (TP14) without antibiotics (n=6/TP); control SPF mice did not receive antibiotics (n=6/TP). (F) Alpha-diversity analysis shown as richness counts and (G) Shannon effective counts. (H) Multidimensional scaling showing differences in the phylogenetic makeup of microbiota between samples (beta-diversity) based on general UniFrac distances. (I) Composition of major phylas determined in cecum content. (J) Portal vein SCFA levels. (K) Levels of selected MUFA, PUFA and MUPC in liver and (L) plasma. AB, antibiotics; Ac, acetate; VM, vancomycin in combination with metronidazole. Red TP in (I-L) indicate a significant difference (p<0.05) between the control and VM group.” (Figure Legends, p.26/27)

- **“Supplementary Figure 4:** Quantitative FA analyses of liver and plasma from mice fed specific diets.

GF and SPF mice (n=6/group) were fed an experimental control diet containing purified cellulose (5%) as fiber source for 2 weeks. (A) Total FA in liver and (B) plasma.

SPF mice (n=6/group) were fed either a chow diet (5% crude fiber) or a regular diet comparable to chow but with elevated fiber content (14% crude fiber) for 2 weeks. (C) Total FA in liver and (D) plasma. Panels (I) Significance and log₂ fold change in individual FA

species; (II) Concentrations of saturated (SA), monounsaturated (MU), and polyunsaturated (PU) FA species shown in boxplots, (III) Individual FA levels of the different groups. Ch, chow; FDR, false discovery rate; GF, germfree; HFi, High Fiber; SPF, specific pathogen-free. * $p < 0.05$, ** $p < 0.01$, *** $p < 0.001$." (Supplementary Figure Legends, p.28/29)

- **“In vivo stable isotope labelling experiments**

SPF C57BL/6N mice (chow diet, 11 weeks of age) were fasted for two hours before application of 0.008g, 0.040g and 0.100g $1\text{-}^{13}\text{C}$ -acetate dissolved in 0.2ml NaCl solution (0.9%) via oral gavage. Mice were sacrificed 4h after oral gavage.” (Methods, p.31)

- **“Dietary intervention experiments**

To study the effects of non-degradable dietary fiber, SPF and GF C57BL/6N mice were fed at the age of 6 weeks a purified control diet with comparable carbohydrate, fat and protein content as chow but with purified cellulose (5%) as fiber source (autoclaved, S5745-E702, Ssniff) for 2 weeks. To study the effects of an altered dietary fiber content, SPF C57BL/6N mice were fed at the age of 6 weeks either a standard chow diet (5% grain-soybean-based crude fiber extract; V1534, Ssniff) or control diet comparable to chow, but with elevated content of degradable fiber (14% grain-soybean-based crude fiber extract; V1574, Ssniff) for 2 weeks.” (Methods, p.33/34)”

- **“Mass isotopomer distribution analysis (MIDA) and fractional synthesis of FA 16:0**

Enrichment of $1\text{-}^{13}\text{C}$ -FA 2:0 in FA 16:0 was analyzed by mass isotopomer distribution analysis using single ion monitoring of molecular ions (M: m/z 270; $M_1\text{-}M_6$: m/z 271-278). Fractional syntheses representing the fraction of newly synthesized FA 16:0 after 4h were calculated by mass isotopomer distribution analysis (MIDA) using the excess mass ratio of the isotopologues M_3/M_2 relative to the unlabeled control (Ecker and Liebisch, 2014; Hellerstein and Neese, 1999).” (Methods, p.37)”

- **“Short chain fatty acids**

Short chain fatty acids (SCFA) were quantified by LC-MS/MS after derivatization to their 3-nitrophenylhydrazones as described by Han et al. (Han et al., 2015). In brief, 10 μl of an internal standard mixture containing 20 $\mu\text{g/ml}$ each D_4 -acetic acid, D_5 -propionic acid, D_7 -butyric acid (Cambridge Isotope Laboratories, USA) were added to 10 μl of portal vein plasma and mixed. 100 μl acetonitrile were added and centrifuged after mixing. 50 μl of the supernatant were derivatized for 30 min at 40°C with each 20 μl of 200 mM 3-nitrophenylhydrazine hydrochloride and 120 mM N-(3-dimethylaminopropyl)-N'-ethylcarbodiimide hydrochloride. The reaction was quenched by addition of 200 μl 0.1 % formic acid. LC separation was performed using a Kinetex® 2.6 μm XB-C18, 50 x 2.1 mm (Phenomenex, Torrance, CA, US) with water as mobile phase A and acetonitrile as mobile phase B both containing 0.1 % formic acid. Gradient elution started with 90 % A with a linear

increase to 81 % A at 0.3 min follow by an increase to 78 % A at 2.5 min. Column was cleaned with 100 % B from 2.6 to 3.0 min and re-equilibrated from 3.1 to 4 min with 90 % A. The column flow was 500 µl at 60°C and 10 µl sample were injected. The method included acetic acid, propionic acid, butyric acid and iso-butyric acid. Butyric acid and iso-butyric acid were separated by LC. For quantification a calibration lines were generated by addition of SCFA to human plasma." (Methods, p.37/38)"

Reviewer I

Comment 2:

“The experimental setting is straightforward, but with limited information e.g. on the mouse study setting, e.g. (1) was sterility of GF mice regularly checked and how?”

The following was added to the manuscript:

- “Sterility of GF mice was routinely confirmed by culturing and microscopic observation of feces after Gram-staining. In addition, 16S rRNA gene-targeted PCR of GF cecal content was performed at the end of the study.” (Methods, p.33)

Comment 3:

“(2) if non-identical housing conditions (isolators) were used after weaning, could that also be one factor affecting the results?”

The antibiotic experiments confirm that our results are independent of mouse housing (Figure 4, Figure 5). To further support this we analyzed the hepatic and plasma fatty acid profiles of GF and Oligo-MM¹² mice housed under identical conditions (isolators).

The following was added to the manuscript:

- **Figure S13** (see last pages of this document)
- “These results are further supported by the finding that Oligo-MM¹² mice show similar differences of the liver and plasma FA profile if compared to GF mice as SPF mice (Figure S13A, B). Oligo-MM¹² mice are housed in isolators within a gnotobiotic environment (as GF mice), they harbor a community of 12 microbial strains representing members of the major bacterial phyla in the murine gut including *Bacteroidetes* and *Firmicutes* (Brugiroux et al., 2016).” (Results, p.8)
- **“Supplementary Figure 3:** Quantitative FA analyses of liver and plasma samples from GF and Oligo-MM¹² mice.
GF and Oligo-MM¹² mice were fed a chow diet (n=6/group). (A) Total FA in liver and (B) plasma. Panels (I) Significance and log₂ fold change in individual FA species; (II) Concentrations of saturated (SA), monounsaturated (MU), and polyunsaturated (PU) FA species shown in boxplots, (III) Individual FA levels of the different groups. FDR, false discovery rate. GF, germfree; OMM, Oligo-MM¹². *p<0.05, **p<0.01, ***p<0.001.” (Supplementary Figure Legends, p.28)

Comment 4:

“The major issue in the study is that too much is concluded based on cross-sectional comparison of two groups of mice in a relatively small study. The observation that fatty acid desaturation and elongation are upregulated in the livers of SPF mice is well supported by the data, but the study does not provide an insight into the potential mechanism behind it.”

A couple of new *in vivo* experiments and analysis have been performed to provide mechanistic insights, please see Comment 1.

Comment 5:

“The antibiotic experiments do suggest that different treatments all seem to produce the GF-like profile, but it would be important to see the effect on lipid metabolism after conventionalisation or colonisation with specific strains, and perhaps with respect to genetic/pharmacological targeting of the pathway of interest. The authors did attempt to analyse the microbial composition after antibiotic treatments, but the results did not offer any insight into their effect on lipid metabolism.”

Data on mice colonized with a defined consortium of 12 strains (Oligo-MM¹²) were included in our manuscript, please see Comment 3.

Moreover, we elaborated that particularly SCFA producers were affected by the antibiotic treatment (Figure 4). The current knowledge is that particularly strains from *Bacteroidetes* and *Firmicutes* generate SCFA including acetate (*Bacteroidetes* > *Firmicutes*). Thus, we eliminated and subsequently recovered *Bacteroidetes* and *Firmicutes* contents in a timeline experiment using a combination of vancomycin and metronidazole (VM). We quantified SCFA, FA and PC species at all time points. Our data show that these phyla can be associated with our lipid metabolic findings identified in the comparison between GF and SPF mice. We think that also the gut microbiota richness (Figure 5F-H) and the total gut bacterial load (Discussion) might play an important role.

The following was added to the manuscript (also listed in Comment 1):

- **Figure 5F-L** (see last pages of this document)
- “Members of *Bacteroidetes* and *Firmicutes* including *Kineothrix alysoides* (OTU-15) degrade dietary polysaccharides from fiber to short chain fatty acids (SCFA) including acetate (FA 2:0), which reach the liver via the portal vein (den Besten et al., 2013; Haas and Blanchard, 2017). Portal blood FA 2:0 levels are ~ 0.5 mM (FA 2:0/FA 3:0/FA 4:0: ~ 50/3/1) in mice, rats and humans (Cummings et al., 1987; Jakobsdottir et al., 2013).“ (Results, p.10)
- “To demonstrate that differences in the lipid profiles of GF and colonized mice (Figure 2, Figure S13), particularly altered MUFA contents, can be associated with microbial FA 2:0 production in the gut, we next manipulated *Bacteroidetes* and *Firmicutes* as major SCFA producers. SPF mice received a combination of vancomycin and metronidazole (VM; as

describe before) for two days (time-point 2, TP2), before they obtained a regular chow diet without antibiotics for additional two (TP4) or 10 days (TP 14). At TP2+VM gut microbial composition and diversity dropped (α - and β -diversity; Figure 5F, G, H). *Bacteroidetes* and *Firmicutes* were almost completely eliminated (Figure 5I) and portal vein FA 2:0 concentrations were reduced 2-fold (Figure 5J). After removal of the antibiotics (TP4 and TP14), the gut microbial ecosystem recovered and baseline FA 2:0 levels (TP0) were reached. Importantly, alterations of liver and plasma MUFA (16:1 *n*-7, FA 18:1 *n*-9, 18:1 *n*-7), MUPC (PC 34:1, PC 36:1) and PUFA (20:3 *n*-6, 22:6 *n*-3) levels clearly followed this trend (Figure 5K, L). These results indicate a direct relation of liver and plasma FA levels to gut microbial SCFA producers and portal blood FA 2:0 levels.” (Results, p.11)

- “Modulation of *Bacteroidetes* and *Firmicutes* contents and potentially the total gut bacterial load (discussed before) by a short-term antibiotic treatment demonstrated a correlation of FA 2:0 levels to liver MUFA and the PUFA fraction. We chose VM, since this antibiotic combination had most pronounced effects on the composite classification score, SCD1 expression, MUFA and *Bacteroidetes* contents (Figure 4). For the production of SCFA, it is important that the gut microbiota work as a community (den Besten et al., 2013; el-Khoury et al., 1994). Thus, we conclude that alterations in gut microbial diversity significantly contribute to SCFA levels in our study. At TP2+VM specifically the fraction of *Proteobacteria* massively rose (Figure 5I). Previous studies suggested that *Proteobacteria* can be associated with FA 3:0 (but not FA 2:0) fitting very well to the observed changes in portal vein FA 3:0 levels (Figure 5J) (Islam et al., 2011; Liou et al., 2013). At TP4, MUFA and MUPC fractions reached even higher concentrations than at TP0 (Figure 5K, L) suggesting a massively induced FA synthesis after antibiotics displacement to restore the initial FA and PC profile.” (Discussion, p.14)

- **“Figure 5:** Enrichment of gut luminal ¹³C-FA 2:0 in hepatic and plasma FA 16:0; association of gut microbial fiber degraders with portal vein SCFA and long chain FA contents of liver and plasma.

SPF mice were supplemented with ¹³C-FA 2:0 via oral gavage with the indicated amount (n=3/group), samples were taken 4h after gavage. (A) Isotopologue distribution of FA 16:0 in liver and (B) plasma. (C) Fractional alterations (%) of M₀, M₁-M₆ isotopologues relative to unlabeled control in liver and (E) plasma. (D) Fraction of *de novo* synthesized FA 16:0 in liver. SPF mice were treated for 2 days with VM (TP2), additional 2 (TP4) or 10 days (TP14) without antibiotics (n=6/TP); control SPF mice did not receive antibiotics (n=6/TP). (F) Alpha-diversity analysis shown as richness counts and (G) Shannon effective counts. (H) Multidimensional scaling showing differences in the phylogenetic makeup of microbiota between samples (beta-diversity) based on general UniFrac distances. (I) Composition of major phylas determined in cecum content. (J) Portal vein SCFA levels. (K) Levels of selected MUFA, PUFA and MUPC in liver and (L) plasma. AB, antibiotics; Ac, acetate; VM, vancomycin in combination with

metronidazole. Red TP in (I-L) indicate a significant difference ($p < 0.05$) between the control and VM group.” (Figure Legends, p.26/27)

Reviewer II

Comment 6:

“Gut microbes play an important role in the energy harvest. Does difference of energy conditions between SPF and GF mice, influence to hepatic lipid metabolisms? How about influence of gut absorbed-nutrients and gut microbial metabolites such as short-chain fatty acids? Authors also should perform water soluble metabolomics of portal blood in addition to lipids metabolome of plasma.”

To the best of our knowledge energy balance parameters were not yet monitored in GF mice adequately/precisely. It would be necessary to design custom build metabolic cages fitting into isolators. As suggested, we focused on energy harvest by degradation and fermentation of dietary fiber to SCFA, we quantified SCFA in portal blood of antibiotic treated mice.

The following was added to the manuscript (also listed in Comment 1):

- **Figure 5 A-E, J** (see last pages of this document)
- “To ask whether FA 2:0 originating from the gut lumen is a precursor for synthesis of long chain FA in the liver, mice were supplied with different concentrations of ¹³C-labeled FA 2:0 via oral gavage. After 4 h liver and plasma FA 16:0 were analyzed for ¹³C-enrichment using GC-MS. Administration of ¹³C-FA 2:0 clearly enhanced isotopologues M₂-M₆ of FA 16:0 in a dose-dependent way in both liver and plasma (Figure 5A, B) resulting in an increase of the fractional abundances of M₂-M₆ and decrease of monoisotopic M₀ (Figure 5C, E). ¹³C-Acetate also dose-dependently elevated the fraction of newly synthesized FA 16:0 in the liver 4 h after oral gavage from ~4 to 30 % demonstrating a stimulation of the hepatic FA *de novo* synthesis by gut-derived acetate (Figure 5D).

To demonstrate that differences in the lipid profiles of GF and colonized mice (Figure 2, Figure SI3), particularly altered MUFA contents, can be associated with microbial FA 2:0 production in the gut, we next manipulated *Bacteroidetes* and *Firmicutes* as major SCFA producers. SPF mice received a combination of vancomycin and metronidazole (VM; as describe before) for two days (time-point 2, TP2), before they obtained a regular chow diet without antibiotics for additional two (TP4) or 10 days (TP 14). At TP2+VM gut microbial composition and diversity dropped (α - and β -diversity; Figure 5F, G, H). *Bacteroidetes* and *Firmicutes* were almost completely eliminated (Figure 5I) and portal vein FA 2:0 concentrations were reduced 2-fold (Figure 5J). After removal of the antibiotics (TP4 and TP14), the gut microbial ecosystem recovered and baseline FA 2:0 levels (TP0) were reached. Importantly, alterations of liver and plasma MUFA (16:1 *n*-7, FA 18:1 *n*-9, 18:1 *n*-7), MUPC (PC 34:1, PC 36:1) and PUFA (20:3 *n*-6, 22:6 *n*-3) levels clearly followed this trend (Figure 5K, L). These results indicate a direct relation of liver and plasma FA levels to gut microbial SCFA producers and portal blood FA 2:0 levels.” (Results, p.10-11)

- “Application of an *in vivo* stable isotope labelling strategy in combination with mass spectrometric analysis revealed that FA 2:0 originating from the gut lumen is used as precursor for hepatic synthesis of FA, which are released into circulation, also as complex lipids. This data are supported by significantly elevated serum lipid levels found in human subjects after rectal infusion of SCFA (Wolever et al., 1989). ¹³C-FA 2:0 doses applied in the present study are within the physiological range of colonic SCFA levels typically found in mice (0.02 g per oral gavage relates to a concentration of ~ 200 mM in the gastrointestinal tract) (Casteleyn et al., 2010; den Besten et al., 2013).

Modulation of *Bacteroidetes* and *Firmicutes* contents and potentially the total gut bacterial load (discussed before) by a short-term antibiotic treatment demonstrated a correlation of FA 2:0 levels to liver MUFA and the PUFA fraction. We chose VM, since this antibiotic combination had most pronounced effects on the composite classification score, SCD1 expression, MUFA and *Bacteroidetes* contents (Figure 4). For the production of SCFA, it is important that the gut microbiota work as a community (den Besten et al., 2013; el-Khoury et al., 1994). Thus, we conclude that alterations in gut microbial diversity significantly contribute to SCFA levels in our study. At TP2+VM specifically the fraction of *Proteobacteria* massively rose (Figure 5I). Previous studies suggested that *Proteobacteria* can be associated with FA 3:0 (but not FA 2:0) fitting very well to the observed changes in portal vein FA 3:0 levels (Figure 5J) (Islam et al., 2011; Liou et al., 2013). At TP4, MUFA and MUPC fractions reached even higher concentrations than at TP0 (Figure 5K, L) suggesting a massively induced FA synthesis after antibiotics displacement to restore the initial FA and PC profile.” (Discussion, p.13/14)

- **“Figure 5:** Enrichment of gut luminal ¹³C-FA 2:0 in hepatic and plasma FA 16:0; association of gut microbial fiber degraders with portal vein SCFA and long chain FA contents of liver and plasma.

SPF mice were supplemented with ¹³C-FA 2:0 via oral gavage with the indicated amount (n=3/group), samples were taken 4h after gavage. (A) Isotopologue distribution of FA 16:0 in liver and (B) plasma. (C) Fractional alterations (%) of M₀, M₁-M₆ isotopologues relative to unlabeled control in liver and (E) plasma. (D) Fraction of *de novo* synthesized FA 16:0 in liver. SPF mice were treated for 2 days with VM (TP2), additional 2 (TP4) or 10 days (TP14) without antibiotics (n=6/TP); control SPF mice did not receive antibiotics (n=6/TP). (F) Alpha-diversity analysis shown as richness counts and (G) Shannon effective counts. (H) Multidimensional scaling showing differences in the phylogenetic makeup of microbiota between samples (beta-diversity) based on general UniFrac distances. (I) Composition of major phylas determined in cecum content. (J) Portal vein SCFA levels. (K) Levels of selected MUFA, PUFA and MUPC in liver and (L) plasma. AB, antibiotics; Ac, acetate; VM, vancomycin in combination with metronidazole. Red TP in (I-L) indicate a significant difference (p<0.05) between the control and VM group.” (Figure Legends, p.26/27)

- **“In vivo stable isotope labelling experiments**

SPF C57BL/6N mice (chow diet, 11 weeks of age) were fasted for two hours before application of 0.008g, 0.040g and 0.100g 1-¹³C-acetate dissolved in 0.2ml NaCl solution (0.9%) via oral gavage. Mice were sacrificed 4h after oral gavage.” (Methods, p.31)

- **“Mass isotopomer distribution analysis (MIDA) and fractional synthesis of FA 16:0**

Enrichment of 1-¹³C-FA 2:0 in FA 16:0 was analyzed by mass isotopomer distribution analysis using single ion monitoring of molecular ions (M: *m/z* 270; M₁-M₈: *m/z* 271-278). Fractional syntheses representing the fraction of newly synthesized FA 16:0 after 4h were calculated by mass isotopomer distribution analysis (MIDA) using the excess mass ratio of the isotopologues M₃/M₂ relative to the unlabeled control (Ecker and Liebisch, 2014; Hellerstein and Neese, 1999).” (Methods, p.37)”

- **“Short chain fatty acids**

Short chain fatty acids (SCFA) were quantified by LC-MS/MS after derivatization to their 3-nitrophenylhydrazones as described by Han et al. (Han et al., 2015). In brief, 10 µl of an internal standard mixture containing 20 µg/ml each D₄-acetic acid, D₅-propionic acid, D₇-butyric acid (Cambridge Isotope Laboratories, USA) were added to 10 µl of portal vein plasma and mixed. 100 µl acetonitrile were added and centrifuged after mixing. 50 µl of the supernatant were derivatized for 30 min at 40°C with each 20 µl of 200 mM 3-nitrophenylhydrazine hydrochloride and 120 mM N-(3-dimethylaminopropyl)-N'-ethylcarbodiimide hydrochloride. The reaction was quenched by addition of 200 µl 0.1 % formic acid. LC separation was performed using a Kinetex® 2.6 µm XB-C18, 50 x 2.1 mm (Phenomenex, Torrance, CA, US) with water as mobile phase A and acetonitrile as mobile phase B both containing 0.1 % formic acid. Gradient elution started with 90 % A with a linear increase to 81 % A at 0.3 min follow by an increase to 78 % A at 2.5 min. Column was cleaned with 100 % B from 2.6 to 3.0 min and re-equilibrated from 3.1 to 4 min with 90 % A. The column flow was 500 µl at 60°C and 10 µl sample were injected. The method included acetic acid, propionic acid, butyric acid and iso-butyric acid. Butyric acid and iso-butyric acid were separated by LC. For quantification a calibration lines were generated by addition of SCFA to human plasma.” (Methods, p.37/38)”

Comment 7:

“Moreover, to clarify direct influence of gut microbes to host hepatic lipid metabolism, authors should examine relationship between host hepatic lipid metabolism and change of gut microbial composition and metabolites on dietary difference (for example; low or high fiber, normal diet or high fat diet) but not between colonized and germ free.”

We fed SPF mice high fiber diet and a chow diet as control, and quantified total FA. It was already previously shown that the dietary fiber contents correlate with gut microbial fiber metabolizers and SCFA levels (Discussion). Because high fat diet might lead to pathophysiologic metabolic consequences, i.e. obesity, this experimental strategy was not applied.

The following was added to the manuscript (also listed in Comment 1):

- **Figure SI4** (see last pages of this document)
- “In agreement, SPF mice fed with control diet containing 14% of fiber had higher levels of MUFA, but lower contents of PUFA compared to SPF mice fed a standard chow diet with 5% fiber (Figure SI4C, D).” (Results, p.11)
- “When SPF mice were fed a diet containing elevated fractions of degradable fiber, gut microbiota-mediated effects were boosted compared to SPF mice fed a regular diet confirming the importance of a degradable fiber source for gut microbiota driven FA synthesis in the liver. It was reported previously that colonic formation of SCFA depends on the amount of dietary fiber ingested (den Besten et al., 2013; Topping and Clifton, 2001).” (Discussion, p.14)
- **“Supplementary Figure 4:** Quantitative FA analyses of liver and plasma from mice fed specific diets.
GF and SPF mice (n=6/group) were fed an experimental control diet containing purified cellulose (5%) as fiber source for 2 weeks. (A) Total FA in liver and (B) plasma.
SPF mice (n=6/group) were fed either a chow diet (5% crude fiber) or a regular diet comparable to chow but with elevated fiber content (14% crude fiber) for 2 weeks. (C) Total FA in liver and (D) plasma. Panels (I) Significance and log₂ fold change in individual FA species; (II) Concentrations of saturated (SA), monounsaturated (MU), and polyunsaturated (PU) FA species shown in boxplots, (III) Individual FA levels of the different groups. Ch, chow; FDR, false discovery rate; GF, germfree; HFi, High Fiber; SPF, specific pathogen-free. *p<0.05, **p<0.01, ***p<0.001.” (Supplementary Figure Legends, p.28/29)

- **“Dietary intervention experiments**

To study the effects of non-degradable dietary fiber, SPF and GF C57BL/6N mice were fed at the age of 6 weeks a purified control diet with comparable carbohydrate, fat and protein content as chow but with purified cellulose (5%) as fiber source (autoclaved, S5745-E702, Ssniff) for 2 weeks. To study the effects of an altered dietary fiber content, SPF C57BL/6N mice were fed at the age of 6 weeks either a standard chow diet (5% grain-soybean-based crude fiber extract; V1534, Ssniff) or control diet comparable to chow, but with elevated content of degradable fiber (14% grain-soybean-based crude fiber extract; V1574, Ssniff) for 2 weeks.” (Methods, p.33)”

Comment 8:

“I think that by integrating these data, authors can find key factors influenced to host hepatic lipid metabolism in gut microbes and metabolites.”

By integrating the suggested data SCFA/acetate was identified as key factor relevant to explain our lipid metabolic findings, see also Comment 1.

GF Oligo-MM¹²

FIGURE S4

REVIEWERS' COMMENTS:

Reviewer #1 (Remarks to the Author):

The manuscript has substantially improved and all of this reviewer's concerns have been adequately addressed and responded to.

Reviewer #2 (Remarks to the Author):

The Authors have adequately responded to Reviewer's comments and concerns. I have no further comments at this point.